# Structural characterization of ribosome recruitment and translocation by type IV IRES

Jason Murray[1,2], Christos G Savva[1], Byung-Sik Shin[2], Thomas E Dever[2], V Ramakrishnan[1]*, Israel S Fernández[1]*†

[1]MRC Laboratory of Molecular Biology, Cambridge, United Kingdom; [2]Laboratory of Gene Regulation and Development, Eunice Kennedy Shriver National Institute of Child Health and Human Development, National Institutes of Health, Bethesda, United States

**Abstract** Viral mRNA sequences with a type IV IRES are able to initiate translation without any host initiation factors. Initial recruitment of the small ribosomal subunit as well as two translocation steps before the first peptidyl transfer are essential for the initiation of translation by these mRNAs. Using electron cryomicroscopy (cryo-EM) we have structurally characterized at high resolution how the Cricket Paralysis Virus Internal Ribosomal Entry Site (CrPV-IRES) binds the small ribosomal subunit (40S) and the translocation intermediate stabilized by elongation factor 2 (eEF2). The CrPV-IRES restricts the otherwise flexible 40S head to a conformation compatible with binding the large ribosomal subunit (60S). Once the 60S is recruited, the binary CrPV-IRES/80S complex oscillates between canonical and rotated states (*Fernández et al., 2014*; *Koh et al., 2014*), as seen for pre-translocation complexes with tRNAs. Elongation factor eEF2 with a GTP analog stabilizes the ribosome-IRES complex in a rotated state with an extra ∼3 degrees of rotation. Key residues in domain IV of eEF2 interact with pseudoknot I (PKI) of the CrPV-IRES stabilizing it in a conformation reminiscent of a hybrid tRNA state. The structure explains how diphthamide, a eukaryotic and archaeal specific post-translational modification of a histidine residue of eEF2, is involved in translocation.

*For correspondence: ramak@mrc-lmb.cam.ac.uk (VR); isf2106@cumc.columbia.edu (ISF)

Present address: †Department of Biochemistry and Molecular Biophysics, Columbia University, New York, United States

Competing interests: The authors declare that no competing interests exist.

## Introduction

During initiation of translation, a crucial step is the placement of the start codon and initiator tRNA in the P site of the ribosome (*Schmeing and Ramakrishnan, 2009*). In eukaryotes, initiation involves almost a dozen protein initiation factors (IFs), several of which are large multi-subunit complexes. A subset of IFs bind to the capped 5' end of a mRNA followed by recruitment of a 43S complex of the small subunit (with additional IFs). Subsequently, the resulting 48S complex scans along the mRNA to reach the start codon (*Aitken and Lorsch, 2012*; *Hinnebusch, 2014*). The complexity of initiation in eukaryotes is used by cells to regulate translation (*Jackson et al., 2010*). However, viruses, which rely on the host machinery to translate their genomes, often target initiation to hijack host ribosomes (*Jackson, 2005*; *Walsh and Mohr, 2011*). One strategy employed by viruses relies on the use of structured sequences in their mRNA that allow them to bypass many or all aspects of canonical initiation (*Pelletier and Sonenberg, 1988*; *Wilson et al., 2000*; *Kieft, 2009*). These structured RNA sequences, termed IRES (for Internal Ribosomal Entry Site) elements, play a critical role in translation of their messages (*Pelletier and Sonenberg, 1988*; *Jackson, 2005*).

Viral IRES sequences can be classified according to their level of dependence on IFs (*Filbin and Kieft, 2009*). Type IV IRESs form a homogenous class of viral RNAs, which completely dispense with

the requirement for any IFs and thus are able to directly recruit and assemble a translating 80S ribosome by themselves. These IRESs are typically found in the intergenic region (IGR) of a family of + sense RNA viruses called *Dicistroviridae* hence they are also known as IGR-IRES (*Wilson et al., 2000*; *Sasaki and Nakashima, 1999*). Extensive biochemical and structural characterization of these IRES sequences has provided a detailed picture on how they work at a molecular level. The approximately 190 nucleotide sequence folds into a compact, well-defined three-dimensional structure making use of three internal pseudoknots (PKI-III, *Figure 1*) (*Costantino and Kieft, 2005*; *Costantino et al., 2008*; *Pestova et al., 2004*; *Schüler et al., 2006*; *Pestova and Hellen, 2003*). Two of these pseudoknots (PKII and PKIII) and accessory sequences, fold into a compact domain following a scheme also seen in other structured RNAs (*Kieft, 2009*; *Au et al., 2015*). As a result, two stem loops (SL-IV and SL-V) are exposed and are responsible for the bulk of interactions with the small ribosomal subunit (*Fernández et al., 2014*; *Schüler et al., 2006*; *Spahn et al., 2004a*). Two additional regions of the IGR-IRES are functionally relevant: the L1.1 region, responsible for the interaction with the L1 stalk of the large ribosomal subunit and PKI. This latter pseudoknot, a structural mimic of a canonical tRNA-mRNA interaction, is responsible for the correct positioning of the viral RNA in the decoding center of the small ribosomal subunit, and is recognized by the ribosome in the same way as a cognate mRNA-tRNA pair (*Figure 1A and D*, *Fernández et al., 2014*; *Costantino et al., 2008*).

Early cryo-EM reconstructions of binary IGR-IRES/ribosome complexes showed how the IGR-IRES is inserted in the intersubunit space of the ribosome, contacting the regions predicted by mutagenesis assays (*Schüler et al., 2006*; *Spahn et al., 2004a*; *Kanamori and Nakashima, 2001*). More recent higher-resolution cryo-EM studies showed that the IRES coexists in equilibrium between canonical and rotated states of the ribosome (*Fernández et al., 2014*; *Koh et al., 2014*). The presence of PKI in the A site is consistent with evidence that translocation of the IRES is required to bring the first codon of the mRNA into the A site and allow delivery of the first animoacyl tRNA (*Yamamoto et al., 2007*; *Zhu et al., 2011*). It also implies that translation of these mRNAs begins in the middle of what is normally the elongation cycle, thus bypassing normal initiation completely and avoiding any regulation over initiation.

Translocation is defined as the concerted movement of tRNAs and mRNA with respect to the ribosomal subunits (*Rodnina and Wintermeyer, 2011*; *Voorhees and Ramakrishnan, 2013*; *Munro et al., 2009*). Upon the acceptance of a cognate tRNA on the A site, peptidyl transfer from the peptidyl tRNA in the P site to the newly accepted aminoacyl tRNA on the A site occurs spontaneously and allows the ribosome to reach an intermediate state of translocation characterized by hybrid A/P and P/E states of the tRNAs and intersubunit rotation (*Moazed and Noller, 1989*; *Frank and Agrawal, 2000*; *Agirrezabala et al., 2008*). This intermediate state is in equilibrium with the original canonical conformation (*Blanchard et al., 2004*). The GTPase elongation factor EF-G in bacteria stabilizes the intermediate state and catalyzes the next step of the process, which involves movement of the mRNA and tRNAs with respect to the small subunit (*Moazed and Noller, 1989*; *Rodnina et al., 1997*; *Spiegel et al., 2007*). Both EF-G and its eukaryotic counterpart eEF2 exhibit a high degree of sequence homology and hydrolyze one molecule of GTP into GDP per translocation cycle. The result of translocation is the movement of the A-site tRNA to the P site and the P-site tRNA to the E site and its subsequent dissociation from the ribosome. Translocation results in an empty A site that can bind the next incoming aminoacyl tRNA. Frameshifting is avoided by coordinate movement of mRNA and tRNAs, which is facilitated by the maintenance of base-pairing interactions between the anticodon stem loops (ASL) of the tRNAs and the mRNA codons during translocation.

Recent structural studies have complemented previous detailed biochemical and single molecule fluorescence studies and shed unprecedented light on the structural nature of various steps of translation elongation and the role of EF-G/eEF2 (*Gao et al., 2009*; *Tourigny et al., 2013*; *Pulk and Cate, 2013*; *Brilot et al., 2013*; *Zhou et al., 2014*). More recently, structures of EF-G/ribosome complexes with both A- and P-site tRNAs have also been determined (*Brilot et al., 2013*; *Zhou et al., 2014*; *Ramrath et al., 2013*). These structures reveal how bacterial EF-G binds to a rotated state of the ribosome, inducing an extra degree of rotation while domain IV of EF-G contacts the tRNA in a hybrid ap/P configuration. In this hybrid state, the tRNA has its anticodon at a position intermediate between the canonical A and P sites in the small subunit while its aminoacyl end is in the P site of the peptidyl transferase center in the large subunit (*Ratje et al., 2010*).

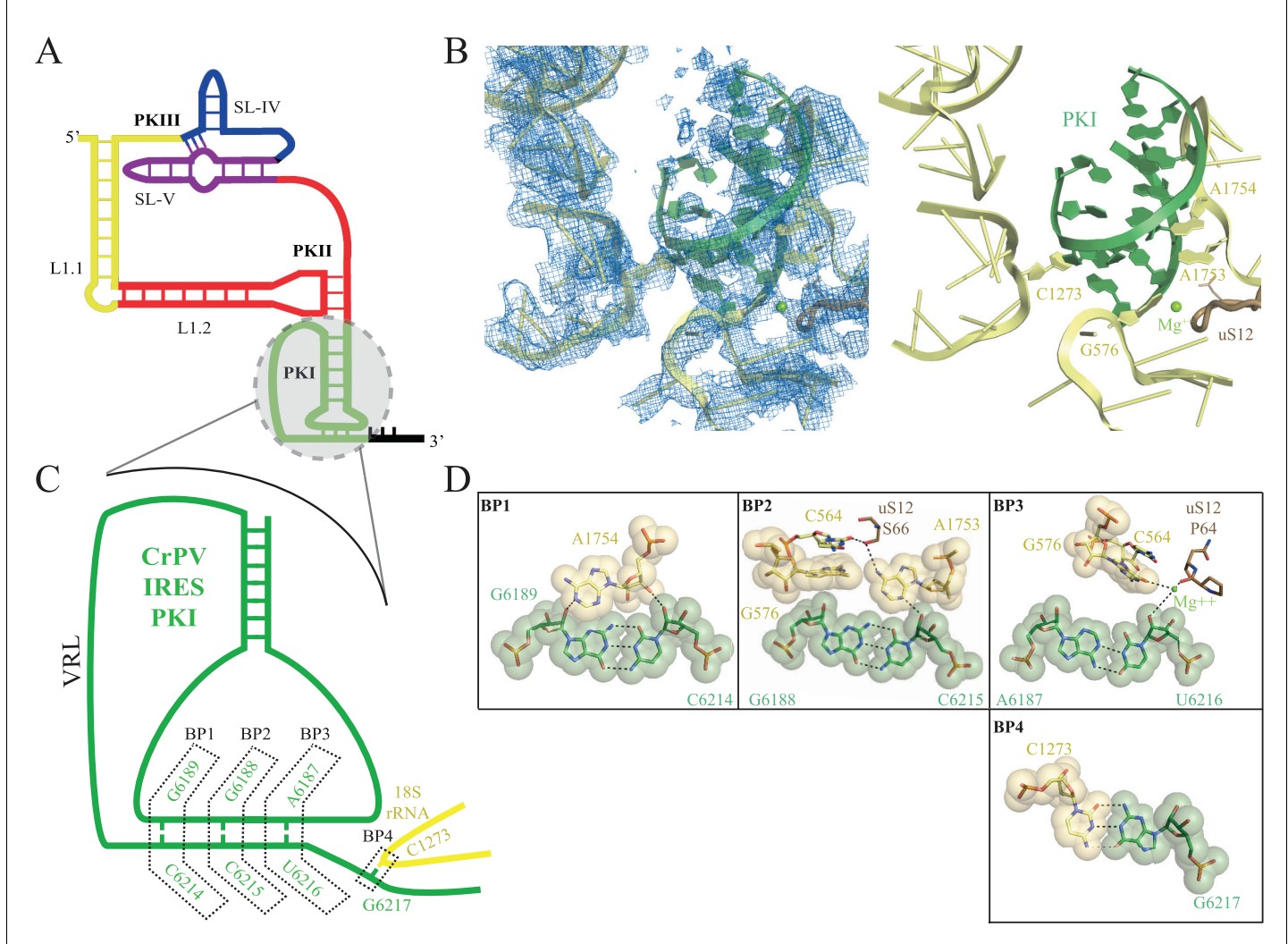

**Figure 1.** Interaction of PKI of CrPV-IRES in the 40S mimics decoding. (**A**) Schematic representation of the type-IV IRES, colored according to different structural motifs. (**B**) Overall view of the interaction of the CrPV-IRES PKI (colored as in (**A**) in the context of the small ribosomal subunit (40S, yellow). On the left, experimental density is shown in the same orientation as on the right, where the final refined model is depicted. (**C**) Schematic view of PKI with main components labelled. (**D**) Space-filling models for the base pairs established between the anticodon and codon mimics of PKI (BP1-3) as well as the adjacent base pair (BP4).

The following figure supplements are available for figure 1:

**Figure supplement 1.** EM image processing workflow for the CrPV-IRES/40S data.

**Figure supplement 2.** Quality of maps and models.

**Figure supplement 3.** Experimental electron density examples.

Even though there are many similarities between bacterial and eukaryotic translocation, some important features are unique to eukaryotes. For example, a conserved histidine residue (H699 in yeast, H719 in mammals) in a loop at the tip of domain IV of eEF2 is post-translationally modified to diphthamide (DPH) (*Schaffrath et al., 2014*). The diphthamide modification, which requires the activity of seven gene products, is conserved in eukaryotes and archaea (*Schaffrath et al., 2014*), but its precise function is unclear. To date, the key feature of diphthamide is that it renders eEF2 susceptible to mono-ADP ribosylation and inactivation by diphtheria toxin, *Pseudomonas* exotoxin

A, and cholera toxin (*Schaffrath et al., 2014*; *Passador and Iglewski, 1994*). Despite uncertainty regarding the cellular function of diphthamide, yeast strains lacking the diphthamide modification enzyme DPH2 display altered levels of frameshifting on the programmed -1 frameshift site from the yeast L-A virus, indicating that diphthamide contributes to fidelity of translation elongation in vivo (*Ortiz et al., 2006*).

Type IV IRES–mediated peptide synthesis requires two prior translocation events: first, movement of the IRES to translocate PKI from the A to the P site, and a second translocation that moves PKI from the P to the E site while simultaneously translocating the first aminoacyl tRNA from the A to the P site (without peptidyl transfer). A precise understanding of how eEF2 works in the context of a ribosome programmed with a type IV IRES is thus needed to fully understand how these IRESs function. Because the IRES mimics tRNAs in the ribosome, such an understanding will also shed light on the role of eEF2 in canonical translocation with tRNAs.

This work addresses two questions. Firstly, it describes the detailed structure of the type IV Cricket Paralysis Virus IRES (CrPV-IRES) bound to the small ribosomal subunit (40S), and how it primes the recruitment of the large subunit in order to assemble a translationally competent 80S. Secondly, the structure of eEF2 with a ribosome-IRES complex in an intermediate state provides insights into eEF2's role in both IRES and canonical translocation.

## Results

### CrPV-IRES interaction with the isolated 40S ribosomal subunit

The first step in translation of an mRNA with a type IV IRES is the recruitment of the small ribosomal subunit (40S) (*Jackson et al., 2010*). To gain insights into the interactions that make this possible, we assembled binary CrPV-IRES/40S complexes for analysis by cryo-EM. Previous data at low resolution had shown the stability of this binary complex (*Spahn et al., 2004a*). A full dataset was collected on a Titan Krios microscope equipped with a Falcon-II direct detector. Given the high degree of compositional as well as conformational heterogeneity present in the sample, extensive classification using a new sorting procedure involving masking and background subtraction (doi:10.1101/025890) was essential to classify homogenous populations (*Figure 1—figure supplement 1*; *Figure 2—figure supplement 1*. A final subset of 54,561 particles produced a homogenous map with an overall resolution of 3.8 Å (*Figure 1*, *Figure 1—figure supplement 1*); however, as previously noted (*Schüler et al., 2006*; *Zhu et al., 2011*), many CrPV-IRES elements have a lower local resolution because they are more dynamic (*Figure 1—figure supplement 2*). Careful analysis of the local resolution (*Kucukelbir et al., 2014*) also showed a high degree of flexibility in regions of the IRES not directly involved in interactions with the 40S, such as L1.1 and L1.2. However, good density could be observed for the functionally relevant regions of the IRES responsible for its binding to the 40S (SL-IV and SL-V) as well as for PKI in the decoding center (*Figure 1—figure supplement 2* and *Videos 2*, *3* and *4* and *Table 1*).

Recent studies of canonical eukaryotic translation initiation suggest an important role for the dynamics of the head of the 40S to correctly position the Met-tRNA$_i$ in the P site (*Llácer et al., 2015*; *Aylett et al., 2015*). Interestingly, the CrPV-IRES appears to have restricted the conformational flexibility of the 40S head. This is mainly accomplished by insertion of SL-IV and V of the IRES in the cleft formed between the head and the body of the 40S, as previously seen in the context of the pre-translocation binary complex CrPV-IRES/80S (*Videos 3* and *4*, *Fernández et al., 2014*).

With this conformation of the head, PKI is able to bind well to the decoding center, allowing for a detailed analysis of its contacts (*Figure 1*, *Figure 1—figure supplement 3* and *Video 2*). While the PKI contacts show marked similarity with canonical decoding, additional features distinguish PKI. Previous mutagenesis studies established the essential role of the correct base pairing of the tRNA/mRNA mimicry domain of type IV IRESs (*Kanamori and Nakashima, 2001*; *Jan and Sarnow, 2002*). Watson-Crick geometry between these bases of PKI is therefore essential to induce a conformation of the decoding center similar to that in decoding with tRNA (*Figure 1B,C and D*, *Ogle and Ramakrishnan, 2005*). Briefly, the conserved adenines 1753 and 1754 (1492 and 1493 in *E. coli*) project outwards from helix 44 (h44) to interact through A-minor motifs with the nucleotides forming base pairs (BP) 1 and 2 of the codon/anticodon minihelix (*Figure 1C*). Accessory elements from other ribosomal components including G576 (G530 in *E. coli*), C564 (C518 in *E. coli*) and residues from uS12 (P64

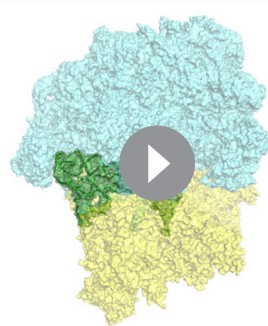

**Video 1.** Video showing the movements of the CrPV-IRES induced by the binding of eEF2 in first translocation event. CrPV-IRES populates an intermediate state between the A site and the P site reminiscent of a ap/P like state. In order to highlight the movement of CrPV-IRES the motions of the 40S ribosomal subunit in relation to the 60S ribosomal subunit have not been shown.

and S66, [P48 and S50 in *E. coli*]) assist in monitoring the shape of BP1 and BP2, and less stringently of BP3 (the wobble position, *Figure 1D and E*). We also observed an additional base pair between position 1 of the first translated codon and the ribosomal base C1273 (C1054 in *E. coli*; termed BP4, *Figure 1B,D and E*). In standard decoding, this ribosomal base interacts with the anticodon nucleotide of BP3, whereas here a rotation of the base about its glycosidic bond allows for a tight stacking interaction with one of the CrPV-IRES bases, redirecting C1273 towards the first base of the first coding codon of the CrPV RNA. All IGR-IRES sequences obtained to date encode a guanine residue at this position with the only exception being the PSIV-IRES (*Kieft, 2009*). This conservation suggests an important role of the additional base pair.

In addition to directing the synthesis of the viral structural protein ORF2, a sub-class of IGR-IRES elements synthetize a second polypeptide called Open Reading Frame x (ORFx), which is encoded in the +1 reading frame relative to ORF2 (*Ren et al., 2012*). Although originally discovered and characterized in Israeli acute paralysis virus (IAPV), Kashmir bee virus (KBV), acute bee paralysis virus (ABPV), and Solenopsis invicta virus (SINV-1), ORFx translation was also recently reported for the CrPV-IRES sequence (*Wang and Jan, 2014*). In order to select an open reading frame, PKI actively assists in presenting the adjacent first codon in a proper conformation in the A site to the first incoming aminoacyl-tRNA. Given the dynamic nature of PKI, its ability to direct initiation of ORF2 or alternatively ORFx could be linked to interactions with ribosomal elements present in the A site. Thus, C1273 may represent an important regulatory element in the ribosomal A site that helps direct frame selection by type IV IRES elements.

In our previous reconstruction of a binary 80S/CrPV-IRES complex, the variable loop (VRL, *Figure 1*) connecting the mRNA-like fragment of PKI with the tRNA-like fragment was poorly ordered (*Fernández et al., 2014*). However, in the context of the 40S subunit, partial density for the VRL can be observed (*Figure 1—figure supplement 3A*). Several studies have explored the structural and functional requirements of the VRL in type IV IRES-mediated translation (*Ren et al., 2014*; *Ruehle et al., 2015*). Interestingly, both the length and sequence of the VRL seem to be essential for the correct positioning of PKI in a productive conformation in the ribosomal A site, as well as for

the translocation steps delivering PKI to the P-site (*Zhu et al., 2011*; *Ruehle et al., 2015*). We speculate the VRL may have an important role in the initial stages of 40S recruitment by type IV IRESs, perhaps contributing additional anchoring points to the small subunit in the initial steps that could help in proper positioning of PKI. Whether this initial placement of PKI (assisted by the VRL) is related to the early events of frame selection by type IV IRESs will require further investigation.

## Overall architecture of a complex of CrPV-IRES with the ribosome and elongation factor eEF2

After recruitment of the large ribosomal subunit (60S), the CrPV-IRES/80S complex oscillates

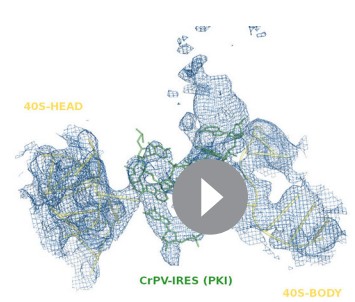

**Video 2.** Video showing experimental density obtained for PKI (green) and the decoding elements of the 40S (yellow) in the CrPV-IRES/40S complex.

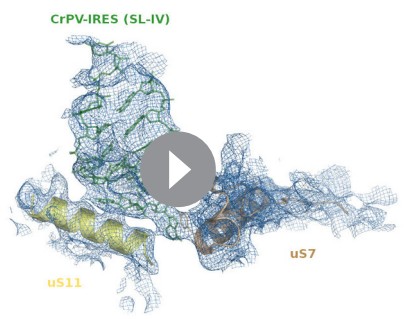

**Video 3.** Video showing experimental density obtained for SL-IV (green) of the CrPV-IRES and ribosomal proteins uS11 (yellow) and uS7 (brown) of the 40S in the CrPV-IRES/40S complex.

between two states of inter-subunit rotation, emulating a canonical pre-translocation complex after peptidyl transfer and prior to translocation (*Fernández et al., 2014*; *Koh et al., 2014*). In this state, PKI remains anchored to the A site of the small subunit and the L1.1 segment couples the rotation of the small subunit with the movement of the L1 stalk of the 60S. In order to allow the first aminoacyl tRNA to be delivered to the A site, PKI has to be translocated to the P site, in a step catalyzed by eEF2. To gain insight into this process, we assembled a pre-translocation complex with CrPV-IRES and 80S, which was subsequently incubated with eEF2 complexed with the non-hydrolysable GTP analog 5'-guanylyl-methylene-bis-phosphonate (GDPCP). This enabled us to trap the pre-translocation complex without the use of any antibiotics.

A large dataset was collected using a FEI Polara microscope equipped with a Falcon-III direct detector. Several rounds of classification in Relion (*Scheres, 2012*) were needed to identify a class of particles consisting of the entire complex that produced a map with an overall resolution of 3.6 Å (*Figure 2*, *Figure 2—figure supplement 1*, *Figure 1—figure supplement 2*). Clear density for eEF2 was visible (*Video 5*), especially for domains G, III, IV and V as well as for the CrPV-IRES (*Figure 1—figure supplement 2D–G*).

The overall conformation of the complex exhibits an additional rotation of the 40S subunit with respect to the 60S of approximately ~3°, when compared with the rotated state previously published without eEF2 (*Figure 2A–B*). Thus, the binding of eEF2 in the GTP form induces an extra rotation of the small subunit as in canonical tRNA translocation (*Brilot et al., 2013*). However, the L1 stalk is 'pushed' in the opposite direction relative to canonical translocation (*Tourigny et al., 2013*), adopting a displaced position that is wider than the widest displacement seen in the rotated form of the CrPV-IRES/80S complex (*Figure 2C*, *Figure 1—figure supplement 3C*, *Video 1*).

## CrPV-IRES conformation with PKI in an ap/P-like state

Translocation is a complex process, involving several orchestrated steps of subunit rotation, tRNA/mRNA movements, and 40S head swivelling (*Wintermeyer et al., 2001*; *Fei et al., 2009*). Structural studies by cryo-EM as well as X-ray crystallography have identified translocation intermediates where the small subunit can be found in several different degrees of rotation along with distinct conformations of the head (*Pulk and Cate, 2013*; *Ratje et al., 2010*). These movements are coupled to the movements of the tRNAs from the A to the P and finally to the E site, with intermediate 'hybrid' states along the way. A useful nomenclature has been coined in order to reflect the diversity of states (*Ratje et al., 2010*). Thus, an ap/P state specifies a tRNA in transit from a canonical A/A state to a canonical P/P state with its anticodon stem-loop (ASL) between the A and P sites in the small subunit, while the CCA-peptidyl end has reached

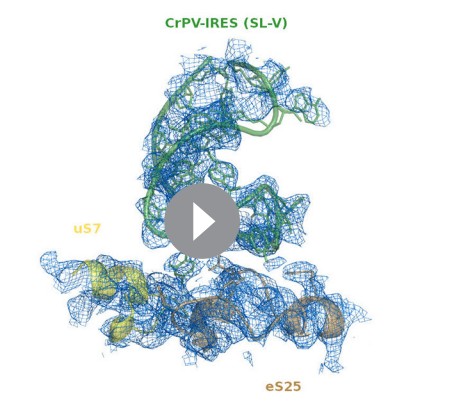

**Video 4.** Video showing experimental density obtained for SL-V (green) of the CrPV-IRES and ribosomal proteins uS7 (yellow) and eS25 (brown) of the 40S in the CrPV-IRES/40S complex.

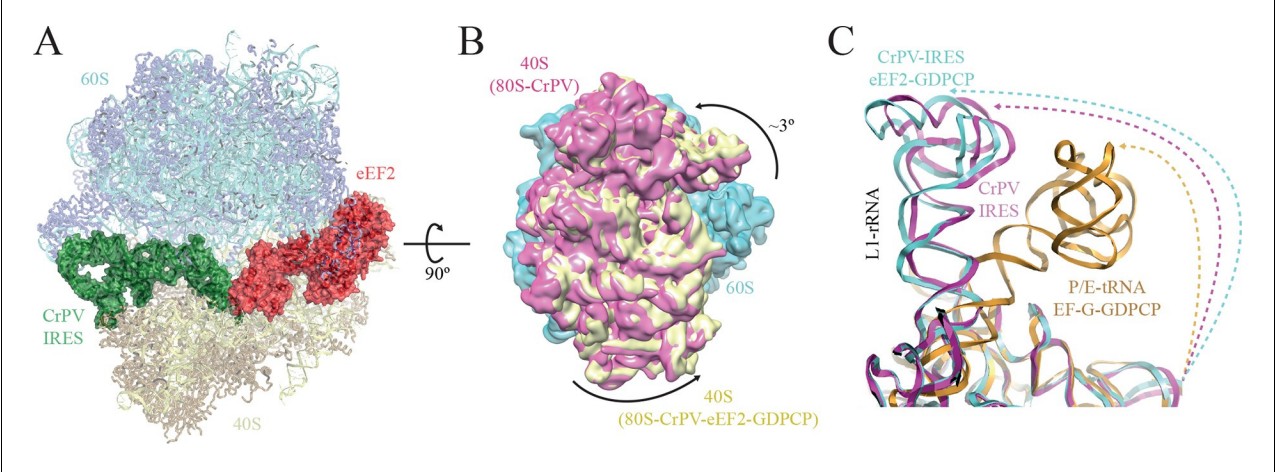

**Figure 2.** Conformational changes in the complex of CrPV-IRES with the 80S ribosome and eEF2. (**A**) Overview of the complex with the large ribosomal subunit (60S) depicted in blue, the small ribosomal subunit (40S) yellow, the CrPV-IRES green and the eEF2 in red. CrPV-IRES and eEF2 are represented as cartoon models inside the corresponding semi-transparent computed molecular surfaces. (**B**) Superposition of low-pass filtered (8Å) volumes from a front view of the 40S showing an additional ~3° rotation induced in the small subunit by eEF2 binding (yellow) when compared with the rotated state without eEF2 (violet, EMDB-2603). No swivelling of the 40S-head was observed. (**C**) Additional displacement of the L1 stalk induced by the binding of eEF2. The RNA component of the L1 stalk in the present structure with eEF2 (blue), the rotated state of the complex without eEF2 (violet; PDB 4V92), and in a complex with tRNAs in the hybrid state (orange; PDB-ID 4V9H).

The following figure supplement is available for figure 2:

**Figure supplement 1.** EM image processing workflow for the 80S/CrPV-IRES/eEF2-GDPCP data.

---

the P-site on the large subunit. This state was described by cryo-EM at ~8 Å resolution (**Brilot et al., 2013**; **Ramrath et al., 2013**). Subsequently a tRNA that is intermediate in both the small and large subunits was seen by X-ray crystallography (**Zhou et al., 2014**). Because PKI mimics an A-site tRNA, the structure here allows us to observe high-resolution details of eEF2 in the A site of the small subunit before translocation.

The overall conformation of the CrPV-IRES in the presence of eEF2-GDPCP is shown in **Figure 3** and **Video 1**. A superposition of the IRES with the positions of tRNAs in their canonical states highlights a displacement of PKI due to the interaction with domain IV of eEF2. The ASL-like element of PKI is pushed into a position in-between the A and P sites of the small subunit, reminiscent of an ap/P state (**Video 1**). Additionally, a superposition of the CrPV-IRES in the ternary CrPV-IRES/eEF2-GDPCP/80S complex with the conformation adopted in the canonical, CrPV-IRES/80S binary complex (excluding PKI in the superposition) reveals a displacement of the ASL-like element of PKI of approximately 10 Å (**Figure 3C**). A comparison of the position of PKI in the CrPV-IRES/40S complex described previously with the position adopted by this pseudoknot in the pre-translocated state, in the CrPV-IRES/eEF2-GDPCP/80S complex described here as well as in the final post-translocated state (**Muhs et al., 2015**) is shown from a top view in **Figure 3D**. In this view, it can clearly be seen that in this inter-mediate, pre-translocated, state, interactions with domain IV of eEF2 induce PKI to adopt an ap/P state.

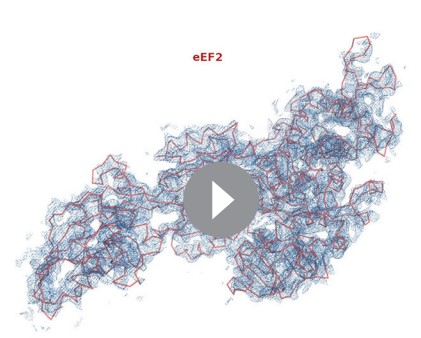

**Video 5.** Video showing experimental density obtained for eEF2 (red ribbon) in the 80S/CrPV-IRES/eEF2 complex.

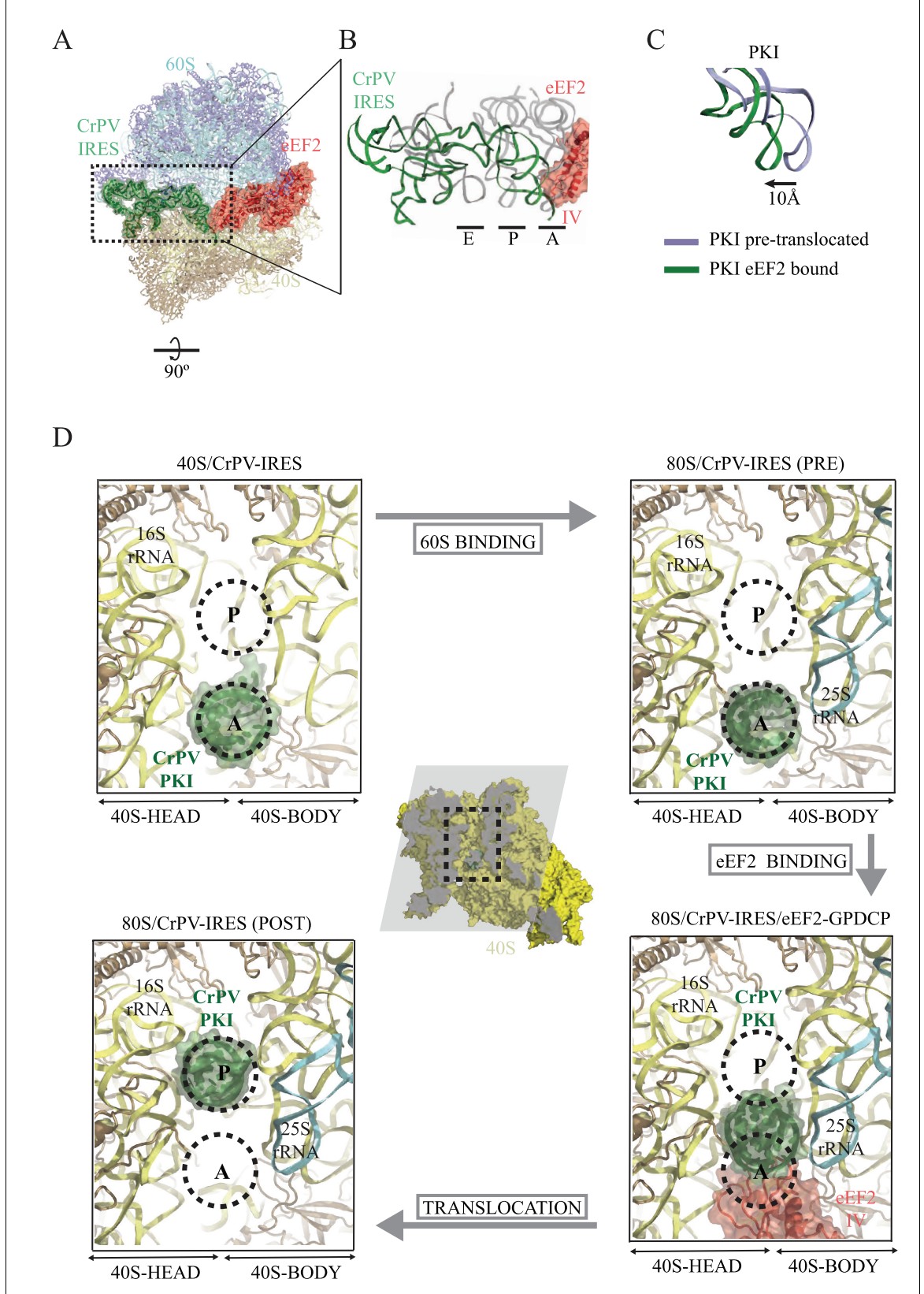

**Figure 3.** CrPV-IRES conformation induced by the binding of eEF2. (**A**) Overview of the complex with the area of interest highlighted. (**B**) Superposition of the CrPV-IRES (green) with tRNAs in canonical configurations (grey, PDB-ID 2WDG). Ribosomal sites A, P and E are indicated to show how the ASL-

*Figure 3 continued on next page*

*Figure 3 continued*

like element of the IRES PKI occupies a position between the A and P sites as it interacts with domain IV of eEF2 (red). (C) Relative displacement of PKI in a comparison of the conformation in the present reconstruction (green) with the one adopted in the CrPV-IRES/80S complex (grey). CrPV-IRES superposition has been computed excluding PKI. (D) View down the axis of the PKI stem in the CrPV-IRES/40S complex (top-left), in the pre-translocated CrPV-IRES/80S complex (top-right *Fernández et al., 2014*), in the 80S/CrPV-IRES/eEF2 complex (bottom-right) and finally, in the post-translocated 80S/CrPV-IRES complex (bottom-left, adapted from *Muhs et al. (2015)*). Ribosomal sites A and P are depicted as dotted lines. The CrPV-IRES PKI (green) is inserted in the A-site in the isolated interaction with the 40S (yellow) as well as in the pre-translocated 80S/CrPV-IRES complex. The interaction with domain IV of eEF2 (red) displaces it towards the P site so it occupies an intermediate position between the A and P sites.

## Interaction of domain IV of eEF2 with PKI and implications for translocation

Domain IV of EF-G is important for its function, since a mutant EF-G lacking the domain is still able to bind to the ribosome, but is unable to catalyze translocation (*Rodnina et al., 1997*). In the present structure, we observed clear density for domain IV of eEF2, and we were able to model its interaction with PKI (*Figure 4*, *Video 5*). Domains III and V of eEF2 solidly anchor the factor to the ribosome through interactions with ribosomal proteins uS12 and uL11 in the 40S and 60S subunits, respectively (*Ban et al., 2014*). These interactions help position eEF2 such that two loops at the tip of domain IV contact PKI (*Figure 4*, *Video 6*). The loop comprising residues 579 to 584 comes close to the major groove of the ASL-like motif of PKI (*Figure 4B–C*). Residues P580 and H583 establish a tight stacking interaction allowing for a sharp turn of the polypeptide chain. At the tip of the bend, K582 is projected outwards to interact with the phosphate of nucleotide 6189 of PKI.

The second loop of eEF2 directly involved in the interaction with PKI encompasses residues 694 to 700 (*Figure 4D*). In our structure, H694 interacts with PKI while the diphthamide moiety (DPH) on H699 is closely packed against the sugar of nucleotide 6189 in the CrPV-IRES. The diphthamide appears to sterically prevent the rRNA decoding bases A1753 and A1754 from interacting with BP1 and BP2 of PKI, whose conformations were otherwise not altered upon binding of eEF2 (*Figure 4E and F*). The interaction between bases A1753 and A1754 and PKI, was seen in the binary CrPV-IRES/40S complex described above, and was maintained during recruitment of the 60S and the induction of the rotated state in the binary CrPV-IRES/80S complex before eEF2 binding. However, this interaction must be broken in order for PKI to translocate from the A to the P site of the 40S. In our structure of the eEF2-bound complex, clear density for these decoding bases was seen in a retracted position in helix 44 ( *Figure 1—figure supplement 3D* ). This suggests that H699, by competing for the binding to PKI with these bases, breaks their decoding interaction with BP1 and BP2. Diphthamide, by making additional stabilizing interactions with PKI facilitates this step. The disruption of the decoding interaction of the ribosomal bases by eEF2 provides an explanation for the well known inhibitory effect of aminoglycoside antibiotics on the action of EF-G and translocation (*Rodnina et al., 1997*; *Studer et al., 2003*). The binding of these antibiotics displaces the ribosomal decoding bases out from helix 44 (h44) (*Carter et al., 2000*; *Ogle et al., 2001*). The antibiotics would thus prevent retraction of the bases into helix 44 upon EF-G or eEF2 binding, thereby precluding interactions of the factors required for their function as well as preventing the disruption of the decoding interactions that is a pre-requisite for translocation (*Figure 4E and F*).

In order to test whether diphthamide plays a role in CrPV-IRES function, we established an *in vitro* reconstituted translation system using reagents from yeast (*Figure 5A*) to monitor IRES-mediated synthesis of the tri-peptide Met-Phe-Lys (MFK). Whereas the CrPV-IRES initiates translation with Ala, a minimal CrPV-IRES RNA, encompassing residues 6027 to 6216, was designed to initiate translation with Met (*Figure 5B*) to facilitate detection of peptide products using [$^{35}$S]Met and to enable direct comparison to peptide synthesis directed by the normal translation initiation pathway. As shown in *Figure 5C*, the WT CrPV-IRES readily directed the synthesis of MFK tripeptide (left panel). Previous studies using mammalian factors (*Jan et al., 2003*) or yeast or mammalian *in vivo* assays (*Wilson et al., 2000*; *Thompson et al., 2001*) established the critical importance of BP1 and BP2 in PKI for IRES-directed translation. Mutation of CC to GG disrupting BP1 and BP2 likewise abolished peptide synthesis in our assays (*Figure 5C*, second panel). As an additional control, a model unstructured mRNA that avoids the requirement for translation factors that promote mRNA recruitment was used to assemble a canonical 80S initiation complex. Notably, assembly of this

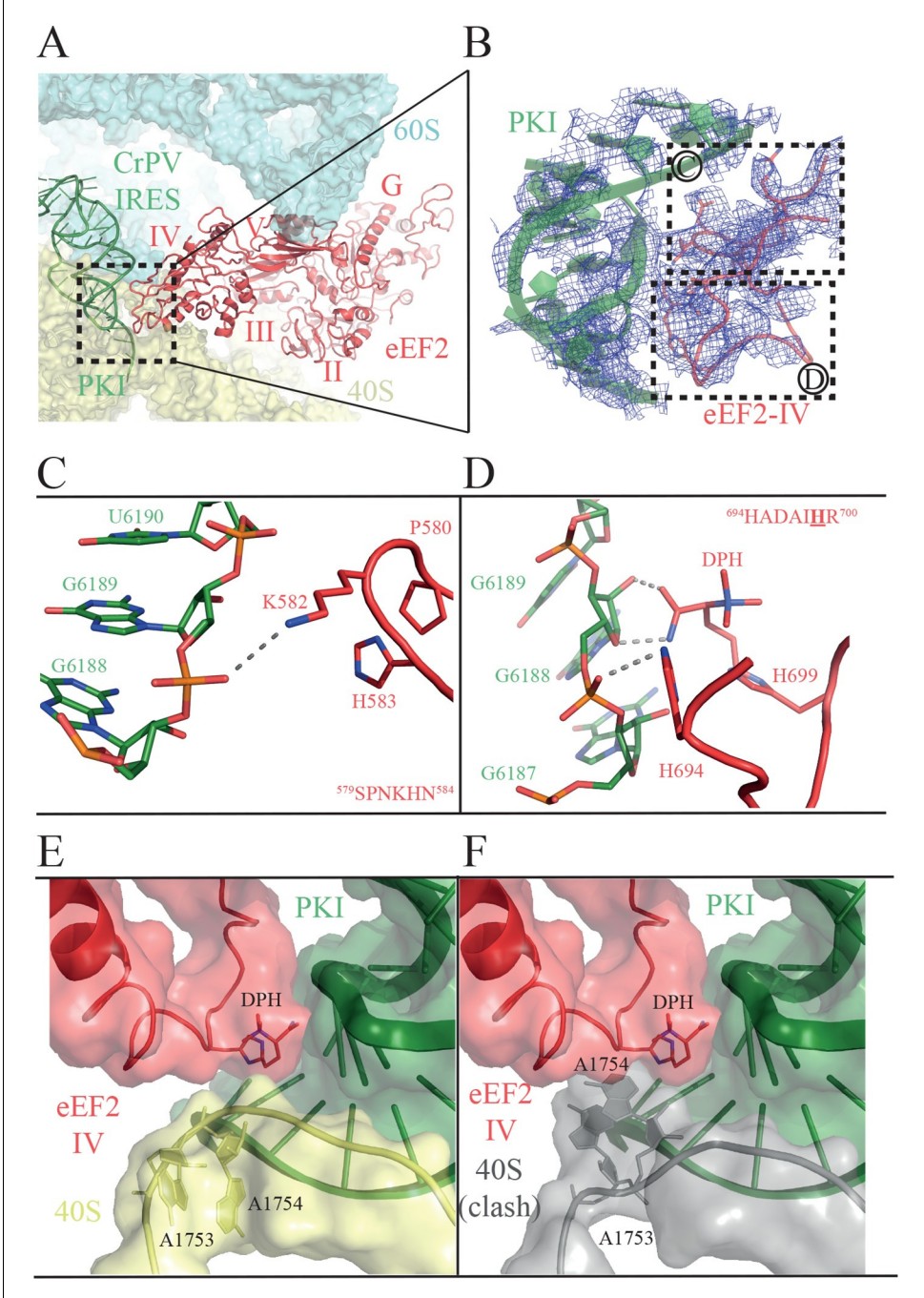

**Figure 4.** Interactions of PKI of CrPV-IRES with domain IV of eEF2. (**A**) Overview of the PKI-eEF2 interaction. The five domains of eEF2 are indicated. (**B**) Experimental density for the two loops at the tip of domain IV of eEF2 interacting with PKI. (**C**) Upper loop of eEF2 (residues 579 to 584) implicated in the interaction with PKI. P580 and H583 of eEF2 closely pack together through a stacking interaction to position K582 within interacting distance of the phosphate group of G6189 of the CrPV-IRES. (**D**) Residues 694–700 in the second loop making direct contacts with PKI. The diphthamide modification of H699 packs against the sugar moiety of G6189. Other residues of this loop such as H694 are also involved in direct contacts with PKI. (**E**) Retracted position adopted by the decoding bases A1753 and A1754 (A1492 and A1493 in *E. coli*) in the CrPV-IRES-eEF2-80S complex. (**F**) The extended conformation of these bases in which they are flipped out of helix 44 as would be seen during decoding or aminoglycoside binding would result in a clash with the tip of eEF2. The small ribosomal subunit components are coloured grey.

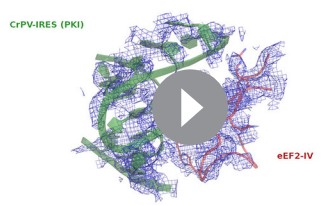

**Video 6.** Video showing experimental density obtained for PKI (green) and domain IV of eEF2 (red) in the 80S/CrPV-IRES/eEF2 complex.

complex required initiation factors eIF1, eIF1A, eIF2, eIF5 and eIF5B (*Gutierrez et al., 2013*). These 80S complexes readily synthesized MFK (*Figure 5C*, fourth panel) with faster kinetics than observed with the CrPV-IRES (*Figure 5C*, right panel). When this same model mRNA was substituted for the CrPV-IRES RNA in reactions lacking initiation factors no peptide synthesis was observed (*Figure 5C*, third panel), confirming the unique property of the IRES to carry out initiation factor-independent translation.

Having established an IRES-dependent in vitro translation system using yeast components, we next examined the impact of eEF2 mutations on CrPV-IRES function. Three eEF2 mutant proteins were purified for these assays. First, eEF2 lacking diphthamide was prepared from yeast lacking the *DPH2* gene required for the first step in diphthamide synthesis (*Schaffrath et al., 2014*; *Ortiz et al., 2006*). Second, eEF2-P580H was designed to disrupt the first loop at the tip of domain IV (see *Figure 4B–C*). Third, eEF2-H699N was designed to eliminate the diphthamide modification of this residue and because previous studies indicated a role for this residue in translation elongation fidelity (*Ortiz et al., 2006*). Consistent with the negligible growth defects under normal conditions in yeast strains lacking *DPH2* or expressing eEF2-H699N (*Ortiz et al., 2006*), the eEF2 mutants functioned like WT eEF2 to promote MFK formation in reactions using 80S initiation complexes (*Figure 5D*). Thus, lack of diphthamide does not significantly impair eEF2 function in normal translation including the translocation events required for MFK formation. In contrast, unmodified eEF2 and the eEF2-P580H and eEF2-H699N mutants exhibited marked defects in MFK synthesis in assays directed by the CrPV-IRES (*Figure 5E*). Since the translocation events prior to the first peptidyl transfer represent key differences between the IRES and canonical pathways for MFK synthesis, diphthamide appears to be important for the non-canonical translocation of the CrPV-IRES prior to first peptide bond formation. The interaction of diphthamide with G6189 of BP1 in the codon-anticodon mimicking element of PKI and the steric obstruction of the rRNA decoding bases by diphthamide suggest that its primary function during normal translation may be to optimize efficiency and fidelity during translocation. Testing this idea will require detailed kinetic studies in a eukaryotic translation system using eEF2 with and without the diphthamide modification.

## A high-resolution view of the eukaryotic GTPase center in an active state

The translational GTPases including eEF2/EF-G, eEF1A/EF-Tu, eIF5B/IF-2, and eRF3/RF-3 assist the ribosome in progressing through the various steps of polypeptide synthesis (*Sprinzl et al., 2000*). All of these factors bind to the ribosome in an active conformation as a complex with GTP. Their binding to the ribosome is followed by hydrolysis of GTP to GDP and subsequent release of the factors from the ribosome. GTP hydrolysis is induced by conformational changes in the ribosome, which position the catalytic site of the factor close to the sarcin-ricin-loop (SRL) in the large subunit. Recent structures of the ribosome in complex with EF-Tu or EF-G bound to GDPCP during decoding or translocation, respectively, have shed light on the molecular details of GTPase activation (*Tourigny et al., 2013*; *Pulk and Cate, 2013*; *Voorhees et al., 2010*). In these structures, three flexible loops (P-loop, switch-I and switch-II) of the GTPase are organized around the GTP (GDPCP) molecule and a highly conserved histidine residue from the switch-II loop (H84 in *E. coli* EF-G) is coordinated to a water molecule close to the γ phosphate (reviewed in [*Voorhees and Ramakrishnan, 2013*]). The conformation of the histidine is stabilized by a universally conserved nucleotide in the sarcin-ricin loop (SRL). Recent biochemical and molecular dynamics studies support the idea that the catalytic center observed in these structures represents a GTPase-activated state (*Maracci et al., 2014*; *Adamczyk and Warshel, 2011*; *Åqvist and Kamerlin, 2015*). However, in structures of both bacterial RF-3 and EF-G with the GTP analog GDPNP, the conserved histidine was observed in a conformation away from both the SRL and the γ phosphate of the GTP molecule

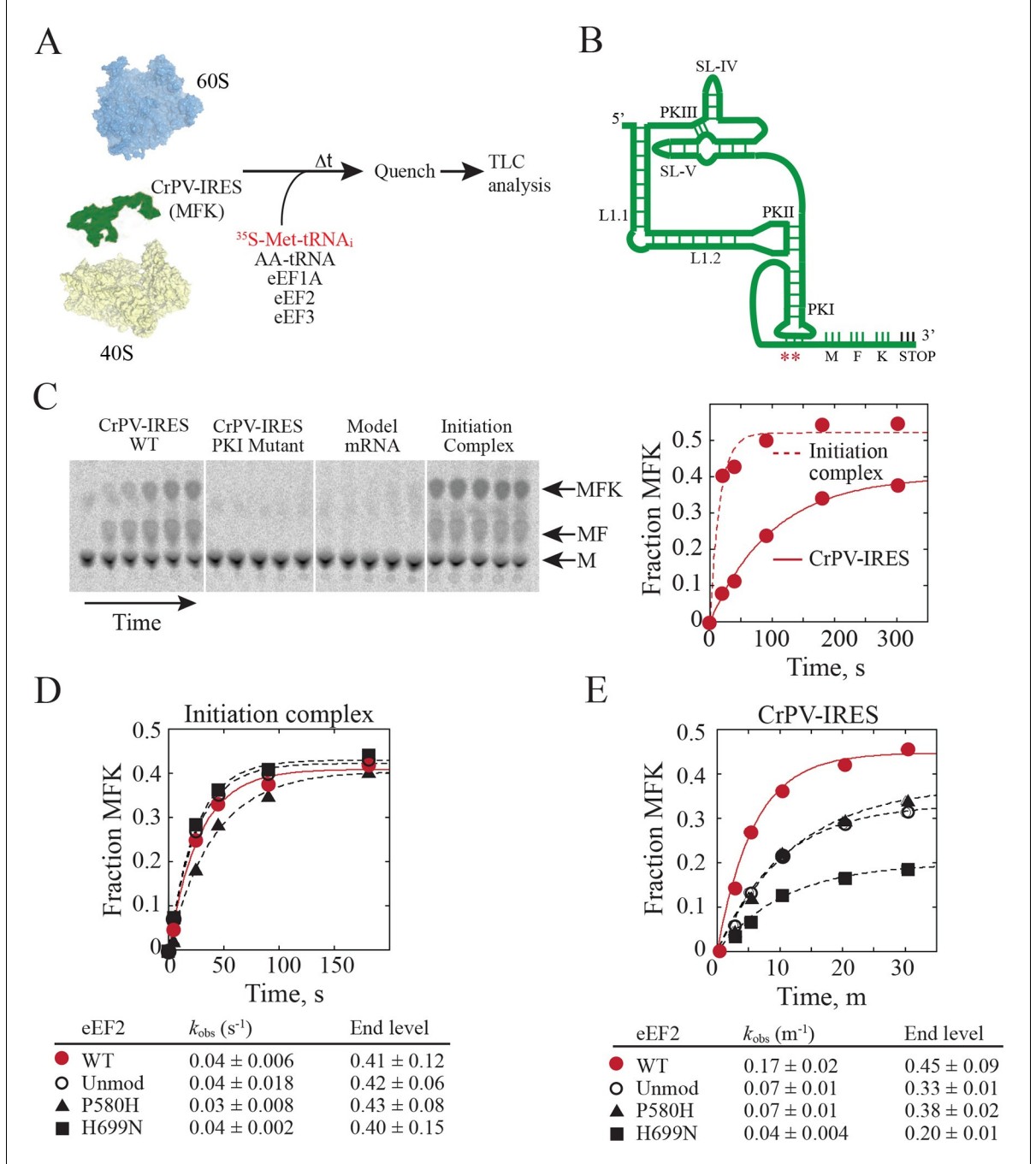

**Figure 5.** Diphthamide modification on eEF2 promotes CrPV-IRES translation. (**A**) Scheme for in vitro reconstituted translation elongation assay with CrPV-IRES RNAs. (**B**) Diagram of the secondary structure of the 200-nt CrPV-IRES. The coding region of the RNA was altered to encode the tripeptide MFK as indicated. Red asterisks denote the site of the CC to GG mutation in BP1 and BP2 of PKI. (**C**) Electrophoretic TLC analysis of peptide products from elongation assays programmed with the CrPV-IRES (left panel), the CC to GG mutant CrPV-IRES (second panel), or a model unstructured 52-nt mRNA encoding MFK (third panel). In the fourth panel, the same 52-nt model mRNA was assembled on 80S initiation complexes using the canonical translation initiation factors. The identities of spots corresponding to free methionine (M), dipeptide MF and tripeptide MFK are indicated. The fraction of MFK synthesis in the reactions with the WT CrPV-IRES and the 80S initiation complex were fit to a single exponential equation and plotted on the right. (**D–E**) Fraction of MFK synthesis in elongation assays programmed with 80S initiation complexes (**D**) or CrPV-IRES (**E**) and using purified WT eEF2, unmodified eEF2 lacking diphthamide (Unmod), eEF2-P580H, or eEF2-H699N, as indicated. Results were plotted and fit to a single exponential equation, and observed rate constants and end levels are shown (note the different time scales for the plots). Errors are standard deviation from three independent experiments.

(*Zhou et al., 2012*; *Zhou et al., 2013*), suggesting that not all GTP analogs behave in the same way and that some may resemble the GTP state less than others.

The GTPase center observed here (*Figure 6*, *Video 7*) agrees with earlier crystallographic results obtained on bacterial systems with GDPCP. Clear density for the P loop, switch I and switch II were visible in our map (*Figure 6A*, *Video 7*) including unambiguous density for the side chain of H108 (H84 in *E. coli*). Additionally, we were able to identify the key magnesium ions described as essential elements for the proposed universal mechanism of GTPase activation (*Figure 6A*, *Video 7*). Although the current structure is of a eukaryotic complex determined by cryoEM, the locations of the key magnesium ions and the overall conformation of the three loops implicated in GTPase activation appear to be conserved with those previously observed in crystallographic studies in bacterial systems (*Figure 6B–C*). This suggests that there is a common and possibly universal mechanism for the activation of translational GTPases by the ribosome, as previously proposed (*Voorhees et al., 2010*).

## Discussion

Type IV IRES sequences are able to fold independently into a compact structure in solution (*Costantino and Kieft, 2005*). Their tertiary structure allows crucial elements of the IRES to interact with the small ribosomal subunit via a cluster of specific ribosomal proteins (eS25, S28, uS7 and uS11) (*Fernández et al., 2014*). This cluster is strategically located in the cleft between the head and body of the 40S. By placing the IRES elements SL-IV and SL-V in that cleft (*Figure 1*), the IRES is able to restrict the otherwise highly dynamic head of the small ribosomal subunit. The restriction on the mobility of the 40S head upon CrPV-IRES binding involves an entropic cost, which is likely balanced by the binding energy from interactions between the CrPV-IRES and the small subunit. This stabilization of the head is accompanied by the insertion of IRES element PKI into the decoding site of the small subunit. The tRNA/mRNA mimicry of this domain induces conformational changes similar to those in decoding, thereby anchoring PKI to the A site. The VRL element of PKI seems to be involved in this early event, as ordered density for a segment of this highly dynamic loop can be distinguished in our reconstruction. The stabilization of the head in an appropriate conformation for 60S binding provides a rationale for how the IRES promotes recruitment of the 60S subunit to form the 80S ribosome. By contrast, those elements of the CrPV-IRES not directly interacting with the 40S (for example, L1.1 and L1.2) are very flexible, and thus, could only be visualized at low resolution in our 40S reconstruction (*Figure 1—figure supplement 2C*). In contrast, these elements, especially L1.1, become ordered upon recruitment of the large ribosomal subunit due to specific contacts, primarily with the L1-stalk (*Fernández et al., 2014*).

The conformation of the 40S head also seems to be restricted by the initiation machinery during delivery of canonical initiator tRNA to the 40S subunit in eukaryotes (*Llácer et al., 2015*; *Erzberger et al., 2014*). Thus, type IV IRESs seem to exploit a feature of ribosomes that is also used in canonical initiation. Other IRESs, such as the type III HCV-IRES, seem to target the same cluster of small subunit ribosomal proteins as the CrPV-IRES (*Quade et al., 2015*), suggesting that this may be a general mode of action of IRES sequences.

PKI is an independent domain of the type IV IRESs that mimics a tRNA anticodon stem-loop interacting with a codon (*Costantino et al., 2008*). This domain is not essential for the binding of the IRES to the small subunit, but it is required for priming the ribosome for elongation (*Jan and Sarnow, 2002*). Previous structural studies in the context of the full (80S) ribosome have unambiguously established the positioning of this PKI in the decoding center in the A site of the small subunit (*Fernández et al., 2014*; *Koh et al., 2014*). Following recruitment of the 60S, the resulting IRES/80S complex oscillates between ratcheted and canonical configurations. This ratcheting of the ribosome involves the classic anti-clockwise rotation of the 40S by approximately 5° (*Figure 7*, top). In this rotated state, which is similar to a canonical pre-translocation complex, the IRES mimics the hybrid states of tRNAs (*Agirrezabala et al., 2008*; *Julian et al., 2008*). During these fluctuations, PKI remains anchored to the decoding center of the A site of the 40S and SL-IV and V remain anchored to the ribosomal protein cluster previously described. GTP hydrolysis by eEF2 catalyzes translocation of PKI to the P site (*Figure 7*, right). Subsequent dissociation of eEF2 results in a vacant A site (*Figure 7*, bottom right) that can accept the first aminoacyl tRNA. A conformation of the IRES with PKI translocated into the P site has recently been characterized using a release factor bound to a stop

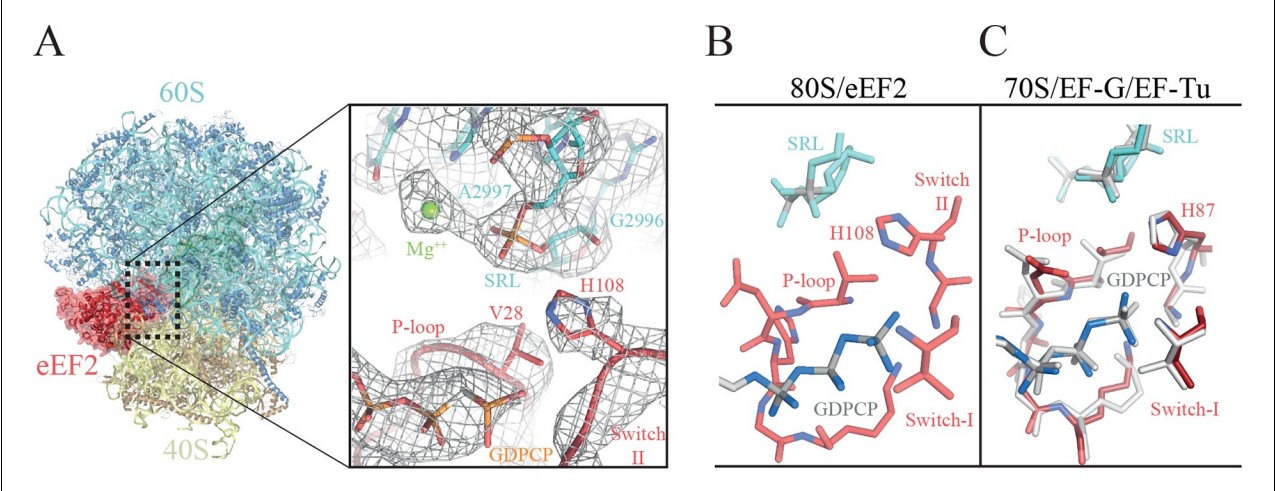

**Figure 6.** The GTPase center of eEF2 is conserved with those of bacterial EF-G and EF-Tu. (**A**) The 80S/CrPV-IRES/eEF2 complex (left) with the inset on the right showing details around the the GTPase centre. Clear experimental density is seen for the GDPCP molecule, the eEF2 residues implicated in catalysis (histidine 108, valine 28), as well as putative magnesium ions (green) that were previously seen with bacterial complexes of GTPase factors. (**B**) The GTPase centre of eEF2 showing the conformation adopted by the P-loop, switch I and II of the eEF2 as well as the ribosomal elements from the sarcin-ricin loop (SRL, cyan). (**C**) Corresponding view of EF-G (colored) (*Tourigny et al., 2013*) and EF-Tu (gray) (*Voorhees et al., 2010*) bound to the bacterial ribosome.

codon in the A site (*Muhs et al., 2015*). In the translocated state, PKI occupies the P site, as expected, but critically, the connections of SL-IV and V with the cluster of ribosomal proteins in the small subunit are lost, leaving these IRES elements solvent exposed. In contrast, the other major anchoring point of the IRES to the ribosome, the L1.1 element (which interacts with the L1-stalk of the 60S), remains solidly in place; however, the stalk is displaced even further compared to its position in the rotated state of the binary IRES-80S complex. The post-translocated conformation of the IRES is characterized by PKI in the P site, SL-IV and V detached from the 40S, and the L1 stalk pushed further away. The entire IRES is rotated around its longitudinal axis when compared to the pre-translocated state (*Figure 7*, bottom-right). Interestingly, in the post-translocated state Muhs *et al* report partial density for the VRL, which is stabilized by interactions with the small subunit (*Muhs et al., 2015*). No density for this loop could be identified in our previous reconstruction of the pre-translocated state (*Fernández et al., 2014*) or in the intermediate state of translocation with eEF2-GPCP presented here. However, partial density could be identified in the CrPV-40S complex. Thus, it seems the VRL alternates between ordered/disordered conformations, possibly regulating the affinities of the compact folds of the IRES, like PKI, as previously suggested (*Ruehle et al., 2015*)

To reach the post-translocated conformation, the IRES has to go through the intermediate state described here, which is stabilized by interaction with eEF2. In the complex with eEF2 and GDPCP, PKI makes a tight interaction with the tip of domain IV of eEF2 and is displaced towards the P site of the ribosome. The ASL-like segment of PKI is in a position and orientation reminiscent of a hybrid ap/P state (*Figure 3*).

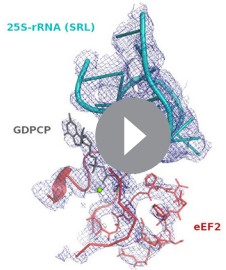

**Video 7.** Video showing experimental density obtained for the GTPase Center in the 80S/CrPV-IRES/eEF2 complex. GDPCP is depicted as grey sticks, eEF2 components are depicted red (switch I as cartoon, switch II as sticks, and P-loop as ribbons) and the sarcin-ricin loop of the 25S rRNA as cyan cartoon. Magnesium ions discussed in the text are represented as green spheres.

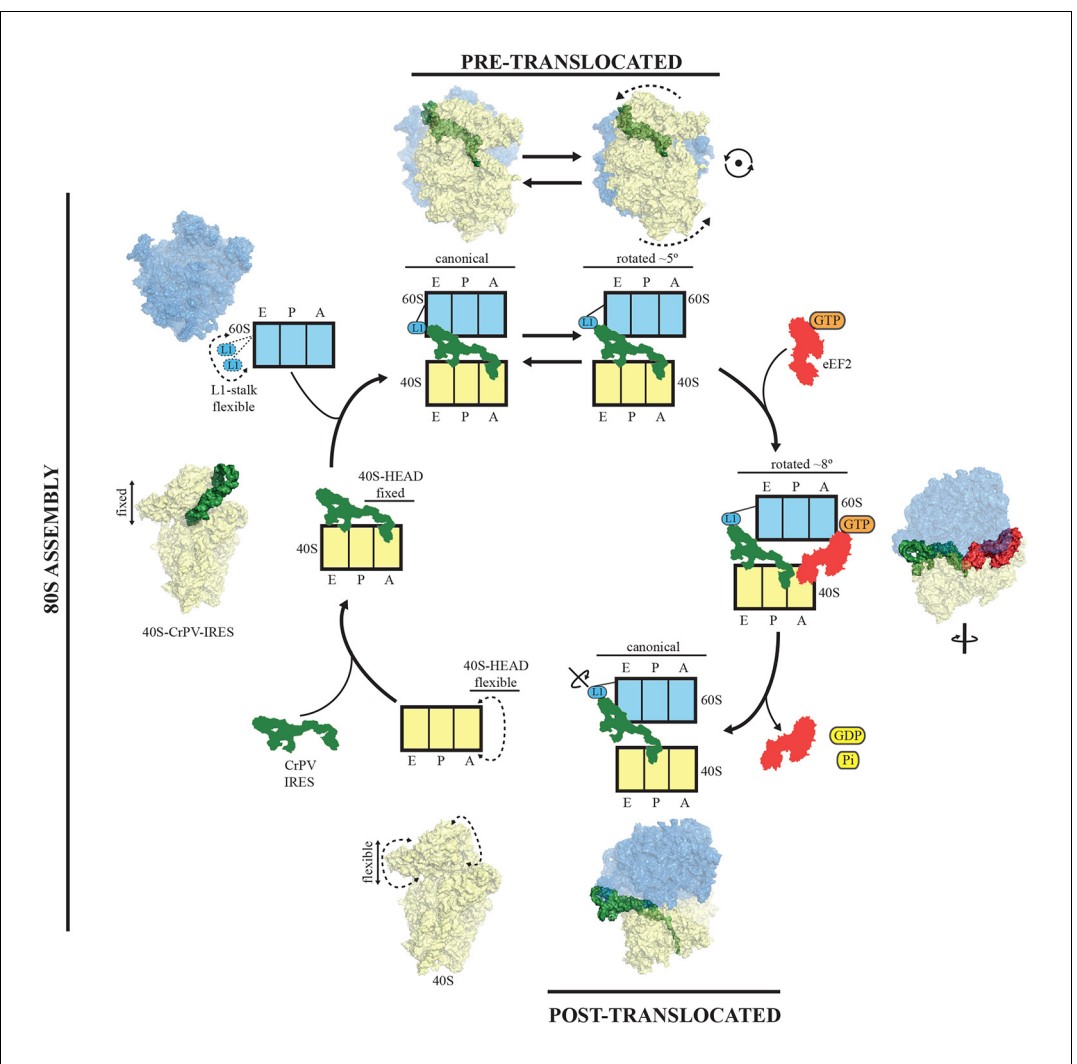

**Figure 7.** A model for initiation of translation by type IV IRESs. Starting from the bottom left, type IV IRESs are able to establish a stable binary interaction with the 40S through the IRES elements SL-IV and V. This interaction restricts the flexibility of the 40S-head in a conformation that both allows decoding of PKI and recruitment of the 60S subunit. Top: once the 60S subunit is recruited, the binary CrPV-IRES/80S complex oscillates between canonical and rotated states mimicking the hybrid states of tRNAs (*Fernández et al., 2014*; *Koh et al., 2014*). Right: eEF2 in its GTP form binds to the rotated state and induces an extra ~3 degree of rotation of the 40S. This extra rotation results in the IRES interacting with the L1 stalk of the 60S in a wider conformation. Domain IV of eEF2 contacts PKI of the IRES and stabilizes it in an intermediate ap/P-like state. Bottom, post-translocated state (*Muhs et al., 2015*): GTP hydrolysis results in conformational changes in eEF2 and the ribosome, the translocation of PKI in the P site, breaking of the original contacts of the IRES elements SL-IV and V with the 40S, and release of eEF2. In order for the IRES to adopt such a conformation, a rotation along its longitudinal axes is necessary.

The whole ribosome adopts a rotated configuration with an additional ~3° anti-clockwise rotation of the small subunit when compared with the rotated state of the binary IRES-80S complex. This extra rotation caused by the binding of eEF2 is also a feature of canonical translocation with tRNAs (*Brilot et al., 2013*). However, unlike the case of canonical translocation, L1 is not stabilized in an inward conformation but instead is displaced outward compared to the binary IRES-80S complex and is very similar to the post-translocated state. Thus, the additional rotation of the 40S subunit, induced by the binding of eEF2 in its active state, forces the IRES to 'push' the L1-stalk to its widest displacement while maintaining the interactions of SL-IV and V with the small subunit. The displaced location of the L1-stalk seems to be compatible with a post-translocated state of the IRES as well as

**Table 1.** Refinement and model statistics for the final refined models.

| Data Collection | CrPV-IRES/40S | CrPV-IRES/eEF2/80S |
|---|---|---|
| Particles | 54,481 | 37,844 |
| Pixel size (Å) | 1.34 | 1.34 |
| Defocus range (µm) | 1.8-3.5 | 1.8–3.5 |
| Voltage (kV) | 300 | 300 |
| Electron dose (e-/Å$^2$) | 25 | 31 |
| **Model composition** | | |
| Non-hydrogen atoms | 79,660 | 205,348 |
| Protein residues | 4,828 | 11,278 |
| RNA bases | 1,781 | 5,264 |
| **Refinement** | | |
| Resolution used for refinement (Å) | 3.8 | 3.6 |
| Map sharpening B-factor (Å$^2$) | -119.8 | -70.8 |
| Average B factor (Å$^2$) | 152.3 | 141.16 |
| MolProbity Clashscore | 97$^{th}$ percentile | 97$^{th}$ percentile |
| MolProbity Score | 90$^{th}$ percentile | 90$^{th}$ percentile |
| Average Fourier Shell Correlation (FSC$_{avg}$) | 0.72 | 0.83 |
| **Rms deviations** | | |
| Bonds (Å) | 0.0057 | 0.0072 |
| Angles (°) | 1.21 | 1.3 |
| **Ramachandran plot** | | |
| Favored (%) | 84.17 | 81.64 |
| Outliers (%) | 4.4 | 4.63 |
| **RNA validation** | | |
| correct sugar puckers (%) | 94.2 | 95.3 |
| good backbone conformations (%) | 55.5 | 65.1 |

with a fully rotated conformation of the 40S in the presence of CrPV-IRES and eEF2-GPDCP. Whether these wide conformations of the L1-stalk are available for normal translocation with tRNAs or they represent a specific state induced solely by IRES sequences remains an open question to be further investigated. Presumably, after GTP hydrolysis and the accompanying conformational changes (namely swiveling of the 40S head plus back-rotation of the 40S subunit (*Ratje et al., 2010*), PKI reaches the P site. Thus, hydrolysis of GTP by eEF2 enables disruption of the interactions of SL-IV and V with the small subunit allowing PKI to reach the P-site. Following this initial translocation, the ribosome is in a state primed for elongation (*Figure 7*).

Our structure of the CrPV-IRES/eEF2-GPDCP/80S complex presents the first images of eEF2 docked on the ribosome with an authentic translocation substrate. Previous cryo-EM studies of yeast eEF2/80S complexes have suggested alternate roles for diphthamide. Based on the structure of an eEF2/80S complex with modeled tRNAs in the A and P sites, Spahn *et al.* proposed that diphthamide was positioned near the codon:anticodon duplex of the A-site tRNA, and that it might functionally replace the decoding bases (*E. coli* A1492/A1493) to stabilize the codon-anticodon pairing during translocation (*Spahn et al., 2004b*). In a subsequent study employing ADP-ribosylated eEF2, it was proposed that domain movements in eEF2 that accompany GTP hydrolysis cause release of the codon:anticodon duplex from the decoding bases during translocation (*Taylor et al., 2007*). More recent structures of eEF2/80S ribosome complexes from humans and *Drosophila* lacked A- and P-site tRNAs and instead contained the yeast Stm1-like serpine 1 mRNA-binding protein 1 (SERBP1) snaking through the mRNA channel and A and P sites of the ribosome. In these latter structures diphthamide adopted two positions: either engaging helix 44 near the decoding bases or

engaging SERBP1 in the mRNA channel (*Anger et al., 2013*). Diphthamide is not essential for yeast cell growth. However, the structure suggests that it additionally stabilizes the interaction of the tip of domain IV with the codon-anticodon interaction, which is presumably maintained during translocation from the A to the P site. This interaction of domain IV would compete with the interaction of the ribosomal decoding bases that also interact with the codon:anticodon duplex in the A site, suggesting how it would facilitate breaking the decoding interactions as a prelude to translocation. A reduction in the strength of the interaction of domain IV with the codon-anticodon helix could explain the enhanced -1 programmed ribosomal frameshifting observed in yeast or mammalian cells when eEF2 lacks the diphthamide modification (*Ortiz et al., 2006*; *Liu et al., 2012*). We propose that the diphthamide modification enhances the fidelity or efficiency of ribosomal translocation and that the unique translocation steps of CrPV-IRES translation, which are uncoupled from peptide synthesis, renders CrPV-IRES translation hypersensitive to the loss of diphthamide (*Figure 5*).

The conformational dynamics of ribosomes play an essential functional role in normal translation. These include the inter-subunit rotation (also known as ratcheting), the swiveling of the head about the long axis of the small subunit, and the fluctuations of the L1-stalk correlated with tRNA in the E site. It has been proposed that these movements are part of the mechanism to harness the thermal energy of the environment and channel it towards the production of useful work (*Spirin, 2009*; *Frank and Gonzalez, 2010*). Type IV IRES sequences are able to use the intrinsic dynamic nature of the ribosome to assemble a translating 80S on their mRNA while avoiding the highly regulated step of normal initiation. We were able to visualize at high resolution an intermediate state of translocation in the CrPV-IRES/eEF2-GDPCP/80S complex. The binding of eEF2 in its active state stabilizes a distorted conformation of the IRES and an additional 3 degrees of rotation of the small subunit. Subsequently, the energy stored in this state could be used along with GTP hydrolysis to enable disruption of interactions present in the initial binding of the IRES to the small subunit, thereby allowing progression of PKI to the P site. The vacated A site is then able to bind the first aminoacyl-tRNA. In conclusion, this work sheds light on various aspects of type IV IRES and eEF2 structure and function during translocation, and provides further evidence for a universal mechanism for GTP hydrolysis by translational GTPases (*Voorhees et al., 2010*).

## Materials and methods

### Preparation of ribosomes

Ribosomes were purified from the *Kluyveromyces lactis* strain GG799 as described previously (*Fernández et al., 2014*). Cells were harvested in mid-log phase ($OD_{600}$ of 2–4) and resuspended in 20 mM MES-KOH pH 6, 150 mM KCl, 150 mM K-acetate, 10 mM $Mg^{+2}$-acetate, 1 mg/ml heparin, 0.1 mM PMSF, 0.1 mM benzamidine, 2 mM DTT. Cell pellets frozen in liquid nitrogen were mechanically disrupted by a blender. The lysate was thawed at 4°C and clarified by centrifugation for 20 min at 14,500 g. Ribosomes in the supernatant were pelleted through a 1M sucrose cushion in the same buffer for 4 hr at 45,000 rpm in a Ti45 rotor (Beckman Coulter, Jersey City, NJ). The pellets were resuspended in 20 mM MES-KOH pH 6, 50 mM KCl, 5 mM $Mg^{+2}$-acetate, 0.1 mM PMSF, 0.1 mM benzamidine, 2 mM DTT, and incubated for 15 min with 1 mM puromycin on ice. The sample was loaded on a 10–40% sucrose gradient in the same resuspension buffer and centrifuged for 16 hrs at 28,000 rpm in a Ti25 zonal rotor (Beckman Coulter, Jersey City, New Jersey). For subunit purification, 80S ribosomes were exchanged into dissociation buffer (20 mM MES-KOH pH 6, 600 mM KCl, 8 mM $Mg^{+2}$-acetate, 1 mg/ml heparin, 0.1 mM PMSF, 0.1 mM benzamidine, 2 mM DTT) before loading onto a sucrose gradient in the same buffer and centrifuged for 19 hr at 28,500 rpm in the Ti25 rotor. The 60S and 40S subunits were exchanged separately into reassociation buffer (10 mM MES-KOH pH 6, 10 mM $NH_4$-acetate, 40 mM K-acetate, 8 mM $Mg^{+2}$-acetate, 2 mM DTT), concentrated to 6 μM and stored at −80°C after being flash frozen in liquid nitrogen.

### Preparation of CrPV IRES

The gene for wild-type CrPV-IRES, was chemically synthesized and cloned in the pUC19 vector flanked at the 5′ by a T7 promoter sequence and an EcoRI cleavage site at the 3′. Standard in vitro transcription protocols were used for the production and purification of IRES RNA.

## Purification of elongation factor eEF2

Elongation factor eEF2 was cloned with a in-frame fused N-terminal calmodulin-binding peptide from genomic DNA extracted from the *Saccharomyces cerevisiae* strain YAS-2488 and introduced into pUC18 vectors containing the expression cassette described previously (*Galej et al., 2013*). These expression cassettes were transferred to pRS424 (eEF2) plasmids. BCY123 cells (*MATα pep4:: HIS3 prb1::LEU2 bar1::HIS6 lys2::GAL1/10-GAL4 can1 ade2 trp1 ura3 his3 leu2-3,112*) containing a target plasmid together with the alternative empty one, were grown on selective medium (lacking uracil and tryptophan) with 1% raffinose to $A600$ nm = 0.8–1.0. Protein expression was induced by the addition of galactose to the final concentration of 2% and cells were grown at 30°C for 12–16 hr. Cell pellets were resuspended in one volume of 2× CAL350 buffer (100 mM Tris-HCl pH 8.5, 500 mM NaCl, 4 mM CaCl2, 2 mM Mg-acetate, 2 mM imidazole, 20 mM 2-mercaptoethanol, 0.2% Igepal CA-630, EDTA-free protease inhibitor cocktail (Roche)), and frozen in liquid nitrogen. Solid-phase cell disruption was performed with freezer mill 6870 (SPEX CertiPrep), the crude extract was adjusted to pH 8 with Tris base and centrifuged at 48,000 $g$ at 4°C for 30 min. The supernatant was incubated with calmodulin-sepharose (recombinant calmodulin coupled to cyanogen bromide-activated sepharose) overnight at 4°C. The resin was washed with CAL500W buffer (20 mM Tris-HCl pH 8, 500 mM NaCl, 2 mM CaCl2, 1 mM Mg-acetate, 1 mM imidazole, 10 mM 2-mercaptoethanol) and eluted with CAL500E buffer (20 mM Tris-HCL pH8.0, 500 mM NaCl, 2 mM EGTA, 1 mM Mg-acetate, 1 mM imidazole, 10 mM 2-mercaptoethanol). After dialysis against Low-Salt-dialysis buffer (20 mM Tris-HCl pH 7.2, 50 mM KCl, 10 mM $NH_4Cl$, 1 mM $MgCl_2$, 2 mM DTT) at 4°C, the samples were further purified by ionic exchange (Q-HP-sepharose colum) and gel filtration (sephacryl-S200). The running buffer for the final gel filtration step was 20 mM Hepes-KOH pH 7.45, 200 mM K-acetate, 10 mM $NH_4Cl$, 1 mM $MgCl_2$, 2 mM DTT). eEF2 eluted from the gel filtration step was concentrated to 3.7 μM, snap frozen in liquid nitrogen and stored at –80°C.

A 5X molar excess of CrPV-IRES (850 nM) was mixed with the 40S subunit (170 nM) in a total volume of 40 μl and incubated for 2 min at 30°C before the sample was cooled to 4°C, diluted to 150 nM 40S, and used immediately to make cryo-EM grids.

For the 80S/CrPV-IRES/eEF2 complex, purified 60S ribosomal subunits (150 nM) were added to the previously described complex followed by a 2 minutes incubation at 30°C. In parallel, purified eEF2 was incubated for 5 minutes at 30°C with 1 mM GPDPC. In a final step, the CrPV-IRES/80S was combined with eEF2/GDPCP for a final incubation of 5 minutes at 30°C and used immediately to make cryo-EM grids.

## Electron microscopy

Aliquots of 3 μl of the CrPV-IRES/40S or CrPV-IRES/eEF2-GPPCP/80S at a concentration of 100–150 nM were incubated for 30 s on glow-discharged holey carbon grids (Quantifoil R2/2), on which a home-made continuous carbon film (estimated to be ~50 Å thick) had previously been deposited. Grids were blotted for 2.5 s and flash cooled in liquid ethane using an FEI Vitrobot. Grids were transferred to an FEI Titan Krios microscope for the CrPV-IRES/40S complex or to a Polara G3 microscope for the CrPV-IRES/eEF2-GPDCP/80S both operated at 300 kV. Defocus values in the final data sets ranged from 1.6–3.6 μm. Images were recorded in automatic mode using EPU software (FEI) on an back-thinned FEI Falcon II detector for the Krios dataset and in a Falcon-III for the Polara dataset at a calibrated magnification of 104478 (yielding a pixel size of 1.34 Å) as described previously (*Bai et al., 2013*). All electron micrographs that showed signs of significant astigmatism or drift were discarded.

## Analysis and structure determination

All reconstructions described were calculated using semi-automated image processing as outlined below. We used the swarm tool in the e2boxer.py program of EMAN2 (*Tang et al., 2007*) for semi-automated particle picking. For our final data sets, we selected 190,895 particles from 1012 micrographs for the 80S/CrPV-IRES/eEF2 complex and 139, 957 particles for the 40S/CrPV-IRES complex from 998 images. Contrast transfer function parameters were estimated using CTFFIND (*Mindell and Grigorieff, 2003*). All 2D and 3D refinements were performed using RELION (*Scheres, 2012*) and the model were manually adjusted to density using COOT (*Emsley et al., 2010*) and refined with REFMAC (*Murshudov et al., 1997*).

## Preparation of aminoacyl-tRNA for peptide formation assay

Yeast initiator tRNA$_i^{Met}$ was prepared by T7 in vitro transcription. As noted previously, in vitro transcribed tRNA$_i^{Met}$ lacking modified nucleotides readily functions in elongation (Astrom and Bystrom, Cell 1994. 79:535–546; Pestova and Hellen 2001. RNA 7:1496–1505; Pestova and Hellen 2003. Genes & Development 17:181–186). Yeast tRNA$^{Phe}$ (Sigma) and tRNA$^{Lys}$ (tRNAprobes, Texas) were obtained from commercial vendors.

The tRNA$_i^{Met}$ was aminoacylated using purified His-tagged *E. coli* MetRS. *E. coli* XL1 Blue cells (Agilent) carrying the MetRS expression plasmid pRA101 (*Alexander et al., 1998*) were grown in 500 ml LB medium containing 100 µg/ml ampicillin at 37°C to A$_{600}$ = 0.5. Following addition of 0.2 mM IPTG the culture was incubated at 20°C for 14 hr. Following harvesting, the cell pellet was suspended in 20 ml lysis buffer (50 mM Tris-HCl [pH 7.5], 300 mM KCl, 6 mM 2-mercarptoethanol, and 10% glycerol) and cells were broken by sonication using a microtip (5 cycles of 30 s pulse followed by 30 s cooling at 4°C). The cell lysate was cleared by centrifugation at 27,000 x g for 30 min and then mixed gently with 1 ml Ni-NTA resin (Qiagen) at 4°C for 1 hr. The resin was transferred to a 1 ml disposable column (Qiagen), washed sequentially with 10 ml lysis buffer, 20 ml lysis buffer containing 20 mM imidazole, and then protein was eluted in 4 ml lysis buffer containing 250 mM imidazole. The elute was dialyzed overnight against 50 mM Tris-HCl (pH 7.5), 25 mM KCl, 10 mM MgCl$_2$, 1 mM DTT and 10% glycerol. For aminoacylation of tRNA$_i^{Met}$, 5 µM tRNA$_i^{Met}$ was mixed with 2 mM ATP, 0.3 µM [$^{35}$S]Met (Perkin Elmer), 10 mM MgCl$_2$ and 1 µM MetRS in reaction buffer (100 mM HEPES-KOH [pH 7.5], 10 mM KCl and 1 mM DTT), and then incubated at 37C for 30 min. Typically, about 60% of tRNA$_i^{Met}$ was aminoacylated with [$^{35}$S]Met.

The tRNA$^{Phe}$ was aminoacylated using purified yeast PheRS. To prepare PheRS, the open reading frames (ORFs) of FRS2 and FRS1, encoding the α and β subunits of yeast PheRS, respectively, were cloned into the polycistronic expression vector pST39 (*Tan, 2001*). First, an N-terminal His$_6$-tagged version of the FRS2 and FRS1 open reading frames were cloned between the *Nde*I and *Bam*HI sites of the vector pET3a Trm, and then FRS2 was moved as an *Xba*I-*Bam*HI fragment to the expression vector pST39 generating the plasmid pC5105. Next, the FRS1 open reading frame was transferred on a *Bsp*EI-*Mlu*I fragment to pC5105 generating the plasmid pC5106. For purification of PheRS, *E. coli* strain BL21(DE3) pLys (Agilent) carrying pC5106 was grown in 500 ml LB medium containing 100 µg/ml ampicillin at 37°C to A$_{600}$ = 0.5. Then, 0.2 mM IPTG was added and the culture was incubated at 25C for 16 hr. Following harvesting, the cell pellet was suspended in 20 ml lysis buffer (20 mM Tris-HCl [pH 7.5], 500 mM KCl, 5 mM MgCl$_2$, 1 mM 2-mercaptoethanol, 10 mM imidazole and 10% glycerol), and PheRS was purified as described above for purification of MetRS. The final PheRS elute was dialyzed overnight against 20 mM Tris-HCl (pH 7.5), 150 mM KOAc, 2.5 mM MgO (Ac)$_2$, 2 mM DTT and 10% glycerol.

To aminoacylate tRNA$^{Lys}$, we first cloned N-terminal His$_6$-tagged KRS1 (Lysyl-tRNA synthetase from *S. cerevisiae*) into pET3a between the *Nde*I and *Mlu*I restriction sites. For purification of LysRS, *E. coli* strain BL21(DE3) CodonPlus-RIL (Agilent) was transformed with pET3a-KRS1 and cells were grown in 250 ml LB medium containing 100 µg/ml ampicillin at 37°C to A$_{600}$ = 0.5. Following addition of 0.5 mM IPTG the culture was incubated at 20°C for 16 hr. LysRS was purified as described above for the purification of PheRS. Aminoacylation of tRNA$_{Phe}$ and tRNA$_{Lys}$ were performed as described previously (*Gutierrez et al., 2013*).

## Peptide formation assay

CrPV IRES RNA (nts 6027–6216) was prepared by T7 in vitro transcription using standard protocol. The DNA template for the T7 in vitro transcription was:

GGCGTAATACGACTCACTATAAGCAAAAATGTGATCTTGCTTGTAAATACAATTTTGAGAGG
TTAATAAATTACAAGTAGTGCTATTTTTGTATTTAGGTTAGCTATTTAGCTTTACGTTCCAGGATGCC
TAGTGGCAGCCCCACAATATCCAGGAAGCCCTCTCTGCGGTTTTTCAGATTAGGTAG
TCGAAAAACCTAAGAAATTTA**CC**T *ATG UUC AAA UAA* UCUCUC; with the T7 promoter underlined, residues 6214 and 6215 that form BP1 and BP2 in PKI and are mutated in the CC to GG mutation (see *Figure 5*) depicted in bold and underlined, and the nucleotides encoding M-F-K-Stop italicized.

Purified ribosomes and translation factors were prepared and in vitro reconstituted peptide formation assays were performed according to procedures described previously (*Gutierrez et al.,*

*2013*) with some modifications. Briefly, 0.4 μM IRES RNA was mixed with 0.4 μM 40S and 60S subunits in 1X reaction buffer (30 mM HEPES-KOH [pH 7.5], 100 mM KOAc, 1 mM MgO(Ac)$_2$, 1 mM spermidine, and 2 mM DTT) and incubated at 26°C for 5 min. An elongation mix (5 nM [$^{35}$S]Met-tRNA$_i^{Met}$, 2 μM eEF1A, 0.4 μM eEF2, 1 μM eEF3, 0.5 μM aminoacyl tRNA, 1 mM GTP-Mg$^{2+}$, and 1 mM ATP-Mg$^{2+}$) in 1X reaction buffer was then added to start the reaction. Reactions were terminated and peptide products were monitored by electrophoretic thin-layer chromatography as described previously (*Gutierrez et al., 2013*). Likewise, peptide formation assays using canonical 80S initiation complexes were performed as described previously (*Gutierrez et al., 2013*). The fractional yield of peptides in each reaction at different times were quantified and fit using Kaleidagraph (Synergy Software) to the single first-order exponential equation $A[1 - \exp(-kt)]$, where A is the amplitude and *k* is observed rate constant.

## Acknowledgements

We are grateful to Shaoxia Chen for technical support with cryo-EM, Greg McMullan for help in video data acquisition, Toby Darling and Jake Grimmett for help with computing, Sjors Scheres, Xiaochen Bai, David Tourigny, Rebecca Voorhees, Alan Brown, Alan Hinnebusch and Jon Lorsch for discussions. JM. thanks the NIH Oxford-Cambridge Scholars' Program for support. This work was funded by grants to VR. from the UK Medical Research Council (MC_U105184332), Wellcome Trust Senior Investigator award (WT096570), the Agouron Institute and the Jeantet Foundation, and in part by the Intramural Research Program of the NIH (TED, JM and B-SS.). The cryo-EM density maps have been deposited in the EMDB with accession numbers EMD-8124 and EMD-8123 (for the CrPV-IRES/40S and CrPV-IRES/eEF2/80S complexes respectively). Atomic coordinates have been deposited in the PDB, with entry codes 5IT9 and 5IT7 (for the CrPV-IRES/40S and CrPV-IRES/eEF2/80S complexes respectively).

## Additional information

### Funding

| Funder | Grant reference number | Author |
|---|---|---|
| Medical Research Council | MC_U105184332 | Jason Murray<br>Christos G Savva<br>Venki Ramakrishnan<br>Israel S Fernández |
| Wellcome Trust | WT096570 | Venki Ramakrishnan<br>Israel S Fernández |
| Agouron Institute | | Venki Ramakrishnan |
| National Institutes of Health | | Jason Murray<br>Thomas E Dever |

The funders had no role in study design, data collection and interpretation, or the decision to submit the work for publication.

### Author contributions

JM, Data acquisition, Structure determination, Manuscript writing, Analysis and interpretation of data; CGS, Help in data acquisition; B-SS, Biochemical experiments, Manuscript writing, Acquisition of data, Analysis and interpretation of data; TED, Experiments design, Manuscript writing, Analysis and interpretation of data; VR, Experiments design, Manuscript drafting; ISF, Design of the experiments, Sample preparation, Structure solution, Manuscript writing, Acquisition of data, Analysis and interpretation of data

### Author ORCIDs

Christos G Savva, http://orcid.org/0000-0003-3354-6483
Israel S Fernández, http://orcid.org/0000-0001-7218-1603

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
