## [Decision Letter]

Thank you for submitting your work entitled "Structural characterization of ribosome recruitment and translocation by a type IV IRES" for consideration by *eLife*. Your article has been favorably evaluated by John Kuriyan as the Senior editor and three reviewers, one of whom, Sriram Subramaniam, is a member of our Board of Reviewing Editors, and another one is Dmitry Lyumkis.

The reviewers have discussed the reviews with one another and the Reviewing Editor has drafted this decision to help you prepare a revised submission.

Summary:

The manuscript "Structural characterization of ribosome recruitment and translocation by a type IV IRES" is a follow-on manuscript to one published from the same research group in 2014 in Cell, titled "Initiation of translation by Cricket Paralysis Virus IRES requires its translocation in the ribosome". In the first manuscript, Fernandez et al. determined the structure of the yeast 80S ribosome bound to the CrPV-IRES, finding that it exists in a dynamic equilibrium between "canonical" and "rotated" conformations. In this second manuscript, Murray et al. determined the structure of the yeast 40S ribosomal subunit bound to CrPV-IRES and the structure of the full 80S complex bound to eukaryotic elongation factor 2 (eEF2). The 80S/CrPV-IRES/eEF2 (with the GTP analog GDPCP) represents a step further along the elongation translocation cycle of the ribosome, just prior to the "post-translocated" state. Overall, the authors have revealed two new structures that offer unique insights into the structure and function of the ribosome during the translocation cycle.

The reviewers raise a number of concerns that must be adequately addressed before the paper can be accepted. It should be possible for you to address these issues without additional experimental work.

Essential revisions:

A major concern is that interpretation of the structures is underdeveloped in light of what is known. What is disappointing is the lack of connecting the structure to mechanistic topics of general interest to the IRES community. But that requires a deeper and more insightful analysis than is present in the manuscript. Consistent with this, there are key references missing (including many recent ones) that must be considered in the analysis. Several examples are given below:

In the last paragraph of the subsection “CrPV-IRES interaction with the isolated 40S ribosomal subunit”. It was not clear what in this analysis of the interactions between the base-pairs of the codon-anticodon interaction and the ribosome is different than what has been described in other structures, including Koh et al. (2014) PNAS 11:9139-44, and previous work by these same authors in Fernandez et al. (2014) Cell 157:823-31. Perhaps this higher resolution work refines those interactions or makes them more certain. If something is truly novel here, the authors need to point it out. Otherwise, focus on the more novel potential findings that may lead to new ideas.

One very interesting and novel finding is the base-pair between C1273 and the first nt in the first decoded codon. But, the authors leave it as perhaps contributing to a "productive conformation in the decoding center." This is not tested experimentally. In fact, the potential of this nt to form a base-pair within the IRES itself has been linked to the ability of some type IV IRESs to initiate in an alternate reading frame, thus this observation may have very exciting implications for IRES regulation (References: Au et al. (2015) PNAS 112:E6446-55; Wang et al. (2014) PloS One 9:e103601; Ren et al. (2012) PNAS 109:E630-9; Ren et al. (2014) Nucleic Acids Res. 42:9366-82.) None of these references, all with direct relationship to this structure, are included. Here and in a few other places, the manuscript (as written) does not reflect the current state of the field.

A missed opportunity: the interactions between the highly conserved apical loops of SL IV and V are known to make critical interactions with eS5 and eS25, but no high-resolution information is known for these interactions. Resolving this key mystery that would be a major contribution. Figure 7/Video 1 – The text and the model in Figure 7 describe SL IV and V as not changing position between the eEF2 unbound to bound state. But, the movie shows something very different. How does the IRES achieve this non-canonical binding and how do the interactions change as the IRES moves? Other than stating an observation about their global position and not much movement occurring, the authors leave this unexplored. The authors miss an opportunity to examine how the Type IV IRESs and possibly other IRESs use these interactions. This is particularly interesting give the literatures regarding eS25 is pretty much as a central player in translation from many IRESs (Thompson lab, in particular).

What is the VLR loop in PKI doing in this mid-translocating state and how does it compare to the pre- and post-translocated state? Video 1 shows some new density appearing at the junction between PKI and PKII upon eEF2 binding. It looks like the authors have modeled it as part of PKI's VLR. It is hard to tell from the movie/any figures. Recent work implicates this loop in elongation factor 2 function – Ruehle et al., (2015) *eLife* –, and it has been explored in manuscripts from the Jan lab. What it is this loop doing and contacting and does it explain this previous work? Are previous predictions about its possible role correct?

In the last paragraph of the subsection “Overall architecture of a complex of CrPV-IRES with the ribosome and elongation factor eEF2”. The position of the L1 stalk is interesting, but this is left with no explanation of potential significance. What might this say about how the IRES works? It is left as just an observation with no context. L1 is well established as a key player in translocation, so what does this odd position suggest? How does this fit into the idea of allosteric communication between L1 and other parts of the ribosome and bound factors? Are eukaryotes just different and what we have learned from bacteria does not apply?

Can the authors say anything about conformational changes within the IRES, outside of PK I? Looking at some modeling from Yamamoto et al., 2007, it would seem the larger part of the IRES may need to move or change structure, a bit for PK I to shift to its new location with the elongation factor. Some discussion of the overall movement of the IRES, and mechanistic implications, is needed.

The position of PKI between the A and P sites was previously suggested or observed in the very first cryo-EM reconstructions of a type IV IRES bound to a ribosome (Spahn Lab). While this was of lower resolution, this previous observation should be acknowledged and discussed. The same is true of the conformational change in the 40S subunit that likely prepares it for 60S subunit binding.

In regard to the observations relating to the diphthamide modification: this is an interesting result and one that bears deeper discussion. The authors suggest that the diphthamide on allows IRES-specific manipulation of the translation machinery that promotes translocation. But the region of the IRES that diphthamide contacts is the *most* similar to a canonical ribosome substrate in the entire IRES, begging the question of why there would need to be an IRES-specific solution there. Furthermore, there is no discussion regarding what is known about how eEF2 contacts canonical tRNA substrates using this modification. Because ribosomal contacts must release the codon:anticodon minihelix in canonical initiation, and they can do so independently of the diphthamide modification (Figure 5), it suggests the ribosome already has mechanisms in place to do this that the IRES should be able to harness as well. If the authors are correct that the modification permits the first noncanonical translocation, would this suggest that there are other, endogenous messages that specifically need to use the diphthamide in a similar way (given its conservation energetic expense for the cell, 8 enzymes)?

Other important points:

The Discussion section seems to largely repeat what was stated in the Results, without adding much insight.

1) Figure 1 – This is essentially the only place where the map of the 40S/CrPV-IRES structure is shown (except for the representation in Figure 1—figure supplement 2 for resolution estimation). The way that it is displayed does not allow the reader to see how complete the structure is, nor how well the model fits to the structure. In D, the authors are obviously using space-filled models for the interactions – actual density for these regions would allow the reader to see if these models actually are supported by the data in those regions.

2) Figure 2 – As with Figure 1, the size and method of displaying the model for the 80D/CrPV-IRES/eEF2 map and model does not allow the reader or reviewer to judge the quality of fit or completeness of the map. Is A the map or model? B appears to be much lower resolution; is this the map? The authors should clearly indicate what they are showing (and should show the density when possible).

3) Figure 3 –This is not well described; the authors need to find a better way to communicate what they are saying here.

4) Figure 4 – As in other figures, when making claims about "clear density" (see subsection “Interaction of domain IV of eEF2 with PKI and implications for translocation”, first paragraph) for side-chains, that clear density needs to be shown. The authors display models in B-D but no density. Also, figures in E and F are unclear: are these models? Density? Of what?

5) Figure 6 – This is the only place in the text where density is shown, but the region being shown is unlabeled. Additional labels in 6A and 6C should be included for clarity. Also, B should show comparison density from that complex as well.

6) The supplemental movie needs more labels if the authors want it to be comprehensible.

7) The reviewers suggest that the manuscript could benefit from tightening the Discussion in some places and expanding it in others. The Introduction is perhaps overly long, several sections of the Results should really have been placed in the Methods section, and a clear presentation of gaps from their previous 2014 paper that the present work fills will make this a more scholarly contribution.

8) The map-to-model validations are incorrect. Why is the FSC between map and model taken at a threshold value of ~0.3 (providing a resolution of ~1/0.26 or ~3.8Å in B and ~1/0.28 or ~3.6Å in F). The correct criterion for the threshold should be 0.5 (see Rosenthal and Henderson 2003). According to the correct criterion, the map-to-model resolution is ~1/0.24 or ~4.2Å in B and ~1/0.26 or 3.8Å in E. Thus, the authors should validate the derived model, for example by performing a cross-validation analysis, similar to that described in Amunts et al. 2014 (from the authors' own laboratory), or by other means.

9) Along the same lines, it would be good to show how well the components of interest correspond to the experimental density, in addition to the full map/model. These could include: (1) local resolution of the CrPV-IRES and eEF2 components; (2) segmented 1/2-map experimental FSC curves and (3) map-to-model FSC curves of CrPV-IRES and eEF2.

---

## [Author Response]

Essential revisions:

*A major concern is that interpretation of the structures is underdeveloped in light of what is known. What is disappointing is the lack of connecting the structure to mechanistic topics of general interest to the IRES community. But that requires a deeper and more insightful analysis than is present in the manuscript. Consistent with this, there are key references missing (including many recent ones) that must be considered in the analysis. Several examples are given below:*

*In the last paragraph of the subsection “CrPV-IRES interaction with the isolated 40S ribosomal subunit”. It was not clear what in this analysis of the interactions between the base-pairs of the codon-anticodon interaction and the ribosome is different than what has been described in other structures, including Koh et al. (2014) PNAS 11:9139-44, and previous work by these same authors in Fernandez et al. (2014) Cell 157:823-31. Perhaps this higher resolution work refines those interactions or makes them more certain. If something is truly novel here, the authors need to point it out. Otherwise, focus on the more novel potential findings that may lead to new ideas.*

Please see our combined responses to the first two comments immediately after the second comment.

*One very interesting and novel finding is the base-pair between C1273 and the first nt in the first decoded codon. But, the authors leave it as perhaps contributing to a "productive conformation in the decoding center." This is not tested experimentally. In fact, the potential of this nt to form a base-pair within the IRES itself has been linked to the ability of some type IV IRESs to initiate in an alternate reading frame, thus this observation may have very exciting implications for IRES regulation (References: Au et al. (2015) PNAS 112:E6446-55; Wang et al. (2014) PloS One 9:e103601; Ren et al. (2012) PNAS 109:E630-9; Ren et al. (2014) Nucleic Acids Res. 42:9366-82.) None of these references, all with direct relationship to this structure, are included. Here and in a few other places, the manuscript (as written) does not reflect the current state of the field.*

We are grateful to the referees in bringing to our attention the relevant references mentioned above. Indeed, extensive mutagenesis and biochemical analysis involving base number one from the first coding codon helps substantially in clarifying the possible mechanism of frame-selection by these types of IRES sequences. We have now added a paragraph in the Results as well as in the Discussion section describing the possible role for the observed base pair established between C1273 and base one of the first codon of the IRES (termed BP4). Relevant references have been included.

*A missed opportunity: the interactions between the highly conserved apical loops of SL IV and V are known to make critical interactions with eS5 and eS25, but no high-resolution information is known for these interactions. Resolving this key mystery that would be a major contribution. Figure 7/Video 1 – The text and the model in Figure 7 describe SL IV and V as not changing position between the eEF2 unbound to bound state. But, the movie shows something very different. How does the IRES achieve this non-canonical binding and how do the interactions change as the IRES moves? Other than stating an observation about their global position and not much movement occurring, the authors leave this unexplored. The authors miss an opportunity to examine how the Type IV IRESs and possibly other IRESs use these interactions. This is particularly interesting give the literatures regarding eS25 is pretty much as a central player in translation from many IRESs (Thompson lab, in particular).*

The overall resolution of the reconstructions are 3.8Å and 3.6Å. However, because the IRES is highly dynamic, the resolution for its elements shows great local resolution variation. We could confidently trace the RNA backbone as well as fitting the PKI domain solved at high resolution by X-ray crystallography (Costantino et al., NSMB, 2008), but the inherent flexibility of the IRES precluded an interpretation of individual interactions of the RNA bases of the IRES with ribosomal proteins. In order to illustrate the quality of the density, several movies with the experimental density for the SL-IV and SL-V as well as the PKI of the IRES in the 40S-complex are contributed as supplementary material (Video 2, Video 3 and Video 4).

What is the VLR loop in PKI doing in this mid-translocating state and how does it compare to the pre- and post-translocated state? Video 1 shows some new density appearing at the junction between PKI and PKII upon eEF2 binding. It looks like the authors have modeled it as part of PKI's VLR. It is hard to tell from the movie/any figures. Recent work implicates this loop in elongation factor 2 function – Ruehle et al., (2015) eLife –, and it has been explored in manuscripts from the Jan lab. What it is this loop doing and contacting and does it explain this previous work? Are previous predictions about its possible role correct?

We thank the reviewers for drawing our attention to the relevant references regarding the VRL. We saw partial density for the VLR in the context of CrPV-IRES/40S reconstruction (new Figure 1—figure supplement 3), which, we speculate, could be related to the early recruitment events of the IRES to the 40S. However, in the reconstruction involving the 80S/CrPV/eEF2 complex the absence of density clearly suggests a disordered VRL at this stage. We added a paragraph in the Results section as well as in the Discussion focusing on how the VLR could play a role in the initial binding of the IRES to the 40S and how our results could be integrated in the previous body of knowledge regarding this specific element of the IRES. Relevant references have also been included.

In the last paragraph of the subsection “Overall architecture of a complex of CrPV-IRES with the ribosome and elongation factor eEF2”. The position of the L1 stalk is interesting, but this is left with no explanation of potential significance. What might this say about how the IRES works? It is left as just an observation with no context. L1 is well established as a key player in translocation, so what does this odd position suggest? How does this fit into the idea of allosteric communication between L1 and other parts of the ribosome and bound factors? Are eukaryotes just different and what we have learned from bacteria does not apply?

A short paragraph has been added in the Discussion section on the putative role of the dynamics of the L1-stalk in type IV IRES-mediated translation. A definitive answer to these questions will require further biochemical and other studies. We hope that this paper will stimulate experimental approaches (like single-molecule FRET) to shed light on the function of this highly mobile element of the ribosome, as has occurred for bacterial translocation.

Can the authors say anything about conformational changes within the IRES, outside of PK I? Looking at some modeling from Yamamoto et al., 2, it would seem the larger part of the IRES may need to move or change structure, a bit for PK I to shift to its new location with the elongation factor. Some discussion of the overall movement of the IRES, and mechanistic implications, is needed.

We saw no additional major conformational changes in the IRES apart from the displacement of the PKI (as shown in Figure 3). A superposition of the RNA backbone of the rest of CrPV-IRES (i.e. excluding PKI) in the pre-translocated state and in the presence of eEF2/GDPCP shows an r.m.s.d of 1.8Å. Thus, there may be subtle changes in the rest of the IRES; however, if they exist, it is not possible to characterize them definitely at this resolution.

*The position of PKI between the A and P sites was previously suggested or observed in the very first cryo-EM reconstructions of a type IV IRES bound to a ribosome (Spahn Lab). While this was of lower resolution, this previous observation should be acknowledged and discussed. The same is true of the conformational change in the 40S subunit that likely prepares it for 60S subunit binding.*

The first visualization of the CrPV-IRES in the context of the 40S and the 80S was achieved in Joachim Frank’s lab (Spahn-Cell-2004) and is appropriately cited in our manuscript. A second, higher resolution, reconstruction for a CrPV/80S complex was reported from the Spahn lab in 2006 (Schuler-NSMB-2006, also cited in our manuscript), which we believe is the reference to which the reviewer is referring. In this 2006 reconstruction, the density for the PKI is highly disordered and the authors placed it “around the P and A sites”. It is important to note that this reconstruction represents a pre-translocated, binary 80S/CrPV-IRES state. In contrast to the Schuler et al. 2006 paper, we and others (Fernandez-Cell-2014, Koh-PNAS-2014) have shown that in the pre-translocated complex, PKI is placed in the A site. It is only upon the action of eEF2 (as in this manuscript) that it occupies a well-defined intermediate state between A and P sites. It is thus our opinion that crediting Schuler-NSMB-2006 as the first reference where PKI was suggested or observed in-between the A and P sites would be misleading. The many other contributions from the Spahn lab are recognized throughout our manuscript, especially the recent structure of a post-translocated state that has been very useful for our analysis of the structures.

In regard to the observations relating to the diphthamide modification: this is an interesting result and one that bears deeper discussion. The authors suggest that the diphthamide on allows IRES-specific manipulation of the translation machinery that promotes translocation. But the region of the IRES that diphthamide contacts is the most similar to a canonical ribosome substrate in the entire IRES, begging the question of why there would need to be an IRES-specific solution there. Furthermore, there is no discussion regarding what is known about how eEF2 contacts canonical tRNA substrates using this modification. Because ribosomal contacts must release the codon:anticodon minihelix in canonical initiation, and they can do so independently of the diphthamide modification (Figure 5), it suggests the ribosome already has mechanisms in place to do this that the IRES should be able to harness as well. If the authors are correct that the modification permits the first noncanonical translocation, would this suggest that there are other, endogenous messages that specifically need to use the diphthamide in a similar way (given its conservation energetic expense for the cell, 8 enzymes)?

We agree that the insights gained from the CrPV-IRES structures has relevance to the function of eEF2 in canonical translation. We propose that diphthamide aids in the similar re-orientation of the decoding bases A1753 and A1754 during canonical translation. While canonical translation can proceed efficiently in the absence of diphthamide, we propose that unique features of PKI and the unusual translocation required for CrPV IRES-mediated translation initiation makes the IRES translation more sensitive than canonical translation to loss of diphthamide. Rather than suggesting, as the reviewers do, that certain cellular mRNAs might use diphthamide in the same way as the CrPV-IRES, we prefer a model in which diphthamide enhances fidelity at all translocation steps and that the unique features of the CrPV-IRES and its translocation confer a heightened requirement for diphthamide. In accord with this line of thinking, we prefer to not speculate that some cellular mRNAs might be translated in a manner similar to the translocation mechanism employed by the CrPV IRES.

Three previous cryo-EM studies have provided modest insights into how diphthamide contacts canonical tRNA translocation substrates. Christian Spahn and Joachim Frank published a low-resolution structure of eEF2 on the 80S ribosome (EMBO J 23:1008-1019 (2004)). Notably, the structure did not contain tRNAs, and instead tRNAs were modeled in. In the model, diphthamide was positioned near the codon:anticodon duplex, and the authors speculated that diphthamide might functionally replace A1492/A1493 to stabilize the codon-anticodon pairing during translocation (they did not consider the idea that diphtamide might help break the decoding interaction). In a follow-up paper by Derek Taylor (EMBO J 26:2421-2431 (2007)), the structure of ADP-ribosylated eEF2 on the 80S was examined. This second paper focused on the impact of GTP hydrolysis by eEF2 and on eEF2 and ribosome conformations. The interaction of eEF2 and diphthamide with tRNA was not discussed; however, it was proposed that movement of the tip of domain IV of eEF2 during GTP hydrolysis might disrupt the interaction of the decoding center with the mRNA-tRNA duplex. Finally, Beckmann and colleagues reported the structure eEF2 bound to *Drosophila* and human ribosomes. Notably, these complexes contained an E-site tRNA and the protein SERBP1 (similar to Stm1 of yeast) snaking through the mRNA channel and the A and P sites of the 80S. The diphthamide moiety on eEF2 adopted two positions in these structures: 1) engaging helix 44 near the decoding bases, or 2) engaging SERBP1 in the mRNA channel. As these structures lack A- and P-site tRNAs, they do not provide insights into how diphthamide interacts with the translocation substrate. Thus, our structure provides the first image of eEF2 and diphthamide interacting with an authentic substrate on the ribosome.

We have edited the text in the Discussion to include these ideas.

Other important points:

The Discussion section seems to largely repeat what was stated in the Results, without adding much insight.

*1) Figure 1 – This is essentially the only place where the map of the 40S/CrPV-IRES structure is shown (except for the representation in Figure 1—figure supplement 2 for resolution estimation). The way that it is displayed does not allow the reader to see how complete the structure is, nor how well the model fits to the structure. In D, the authors are obviously using space-filled models for the interactions – actual density for these regions would allow the reader to see if these models actually are supported by the data in those regions.*

We added a panel with the experimental density for the decoding elements of the 40S in the presence of CrPV-PKI (B, left panel). In order to be consistent with the original high resolution structural paper on decoding (Ogle-Science-2001), we feel it is better to display the base pairs as space-filling models, because they are derived from a stereo-chemically-correct, refined model. However, we changed the figure legend to clearly reflect this and avoid misinterpretation; we also show the experimental density in Figure 1, Figure 1—figure supplement 3 and Video 2.

2) Figure 2 – As with Figure 1, the size and method of displaying the model for the 80D/CrPV-IRES/eEF2 map and model does not allow the reader or reviewer to judge the quality of fit or completeness of the map. Is A the map or model? B appears to be much lower resolution; is this the map? The authors should clearly indicate what they are showing (and should show the density when possible).

We used space-filling models for the CrPV and eEF2 in panel A as has been done in many papers describing structures of ribosomal complexes by X-ray crystallography at similar resolution, either from our group (Neubauer –Cell-2009, Gao-Science-2009) or from other groups (Zhou-Science-2014, Blaha-Science-2009). At this resolution an overall view with density would be very confusing so we decided to clarify in the figure legend what is shown and additionally we now include a movie with the experimental density for eEF2 (Video 5). The density for the L1-stalk is shown in Figure 1—figure supplement 3. Additionally, all relevant computed maps as well as the refined models will be made publicly available.

3) Figure 3 –This is not well described; the authors need to find a better way to communicate what they are saying here.

To make interpretation of the figure easier for the reader, we have reorganized the panels and added connecting information between the panels. Also, labels for each panel have been added to make it easier to identify the components of each view.

To properly explain the positioning of the CrPV-IRES PKI, it was mandatory to isolate PKI from the rest of the model, otherwise the representation was very messy. We feel that using descriptions similar to those used previously to describe tRNAs in several configurations (canonical and hybrid) is a good idea given the structural similarity relating the PKI with a canonical tRNA/mRNA pair. Thus, we have chosen to use this type of representation (top view of the solvent side of the small subunit) in panel D to highlight the occupancy of the ribosomal A and P sites. This mode of representation is frequently used in the ribosome field (e.g. Selmer-Science-2006, Figure 3; Blaha-Science-2009, Figure 5; Zhou-Science-2014, Figure 2).

4) Figure 4 – As in other figures, when making claims about "clear density" (see subsection “Interaction of domain IV of eEF2 with PKI and implications for translocation”, first paragraph) for side-chains, that clear density needs to be shown. The authors display models in B-D but no density. Also, figures in E and F are unclear: are these models? Density? Of what?

We have replaced panel 4B with a close-up view of the area of interest that includes the experimental density. Additionally, we added a short movie with the experimental density for eEF2 (Video 6). We explicitly clarify in the figure legend that panels E-F display Van der Waals rendered volumes of the refined model. These images illustrate the incompatibility of the position of the decoding bases 1753 and 1754 in their outward configuration with the position we observe for the tip of domain IV of eEF2. The steric clash between these eEF2 and the decoding bases is illustrated in panel F where the 40S is depicted in a different color to emphasize the clash. We have expanded the figure legend and added more labels to aid the reader.

*5) Figure 6 – This is the only place in the text where density is shown, but the region being shown is unlabeled. Additional labels in 6A and 6C should be included for clarity. Also, B should show comparison density from that complex as well.*

Labels have been added for panels A and C. The density for panel B is shown in panel A as well as in Video 7. In panel C two superposed complexes are depicted (70S/EF-Tu and 70S/EF-G). Showing density for these two complexes (determined by X-ray crystallography at different resolutions) would be very messy.

6) The supplemental movie needs more labels if the authors want it to be comprehensible.

More labels have been added.

7) The reviewers suggest that the manuscript could benefit from tightening the Discussion in some places and expanding it in others. The Introduction is perhaps overly long, several sections of the Results should really have been placed in the Methods section, and a clear presentation of gaps from their previous 2014 paper that the present work fills will make this a more scholarly contribution.

We have edited the text in several places to address this comment.

8) The map-to-model validations are incorrect. Why is the FSC between map and model taken at a threshold value of ~0.3 (providing a resolution of ~1/0.26 or ~3.8Å in B and ~1/0.28 or ~3.6Å in F). The correct criterion for the threshold should be 0.5 (see Rosenthal and Henderson 2003). According to the correct criterion, the map-to-model resolution is ~1/0.24 or ~4.2Å in B and ~1/0.26 or 3.8Å in E. Thus, the authors should validate the derived model, for example by performing a cross-validation analysis, similar to that described in Amunts et al. 2014 (from the authors' own laboratory), or by other means.

Figure 1—figure supplement 2 shows, for both reconstructions, FSC curves between half-maps independently computed as suggested in Scheres-NatMethods-2012 (the Gold Standard FSC) (A and E). In panels B and F three curves are shown: black line is the model-map FSC using the final map and the final refined model; red and blue lines correspond to the cross-validation analysis, as first proposed in Amunts et al., Science (2014). Perhaps the overlapping of the two curves precluded clear observation, but the requested analysis has been performed and is shown. The superposition of the blue and red curves highlights the absence of over-fitting of the refined model as explained in Brown et al., ActaCrystD (2015). We have expanded the figure legend to make this explicit. In the expanded figure legend we also explain that the vertical line represents the maximum resolution included in the refinement.

9) Along the same lines, it would be good to show how well the components of interest correspond to the experimental density, in addition to the full map/model. These could include: (1) local resolution of the CrPV-IRES and eEF2 components; (2) segmented 1/2-map experimental FSC curves and (3) map-to-model FSC curves of CrPV-IRES and eEF2.

Local resolution maps for the ligands (CrPV-IRES and eEF2) are now shown in the Figure 1—figure supplement 2 (panels C and G).